# Quantifying the Plausibility of Context Reliance in Neural Machine Translation

**Gabriele Sarti**[1]    **Grzegorz Chrupała**[2]    **Malvina Nissim**[1]    **Arianna Bisazza**[1]

[1]Center for Language and Cognition (CLCG), University of Groningen
[2]Dept. of Cognitive Science and Artificial Intelligence (CSAI), Tilburg University
{g.sarti, m.nissim, a.bisazza}@rug.nl, grzegorz@chrupala.me

## Abstract

Establishing whether language models can use contextual information in a human-plausible way is important to ensure their trustworthiness in real-world settings. However, the questions of *when* and *which parts* of the context affect model generations are typically tackled separately, with current plausibility evaluations being practically limited to a handful of artificial benchmarks. To address this, we introduce **P**lausibility **E**valuation of **Co**ntext **Re**liance (PECoRe), an end-to-end interpretability framework designed to quantify context usage in language models' generations. Our approach leverages model internals to (i) contrastively identify context-sensitive target tokens in generated texts and (ii) link them to contextual cues justifying their prediction. We use PECoRe to quantify the plausibility of context-aware machine translation models, comparing model rationales with human annotations across several discourse-level phenomena. Finally, we apply our method to unannotated model translations to identify context-mediated predictions and highlight instances of (im)plausible context usage throughout generation.

## 1 Introduction

Research in NLP interpretability defines various desiderata for rationales of model behaviors, i.e. the contributions of input tokens toward model predictions computed using feature attribution (Madsen et al., 2022). One of such properties is *plausibility*, corresponding to the alignment between model rationales and salient input words identified by human annotators (Jacovi & Goldberg, 2020). Low-plausibility rationales usually occur alongside generalization failures or biased predictions and can be useful to identify cases of models being "right for the wrong reasons" McCoy et al. (2019). However, while plausibility has an intuitive interpretation for classification tasks involving a single prediction, extending this methodology to generative language models (LMs) presents several challenges. First, LMs have a large output space where semantically equivalent tokens (e.g. *"PC"* and *"computer"*) are competing candidates for next-word prediction (Holtzman et al., 2021). Moreover, LMs generations are the product of optimization pressures to ensure independent properties such as semantic relatedness, topical coherence and grammatical correctness, which can hardly be captured by a single attribution score (Yin & Neubig, 2022). Finally, since autoregressive generation involves an iterative prediction process, model rationales could be extracted for every generated token. This raises the issue of *which generated tokens* can have plausible contextual explanations.

Recent attribution techniques for explaining language models incorporate contrastive alternatives to disentangle different aspects of model predictions (e.g. the choice of *"meowing"* over *"screaming"* for *"The cat is ___"* is motivated by semantic appropriateness, but not by grammaticality) (Ferrando et al., 2023; Sarti et al., 2023). However, these studies avoid the issues above by narrowing the evaluation to a single generation step matching a phenomenon of interest. For example, given the sentence *"The pictures of the cat ___"*, a plausible rationale for the prediction of the word *"are"* should reflect the role of *"pictures"* in subject-verb agreement. While this approach can be useful to validate model rationales, it confines plausibility assessment to a small set of handcrafted benchmarks where tokens with plausible explanations are known in advance. Moreover, it risks overlooking important patterns of context usage, including those that do not immediately match linguistic intuitions. In light of this, we suggest that identifying *which* generated tokens were most affected by contextual input information should be an integral part of plausibility evaluation for language generation tasks.

Figure 1: Examples of sentence-level and contextual English→Italian MT. Sentence-level translation contain lack-of-context errors. Instead, in the contextual case context-sensitive source tokens are disambiguated using source (Ⓢ) or target-based (Ⓣ) contextual cues to produce correct context-sensitive target tokens. PECoRe enables the end-to-end extraction of cue-target pairs (e.g. <she, alla pastorella>, <le pecore, le>).

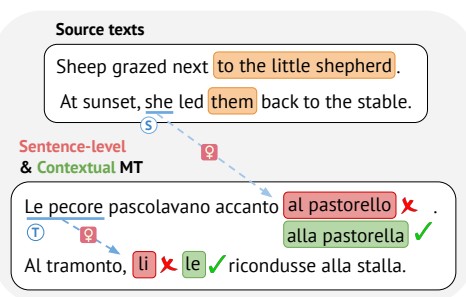

To achieve this goal, we propose a novel interpretability framework, which we dub **P**lausibility **E**valuation of **Co**ntext **Re**liance (PECoRe). PECoRe enables the end-to-end extraction of *cue-target token pairs* consisting of context-sensitive generated tokens and their respective influential contextual cues from language model generations, as shown in Figure 1. These pairs can uncover context dependence in naturally occurring generations and, for cases where human annotations are available, help quantify context usage plausibility in language models. Importantly, our approach is compatible with modern attribution methods using contrastive targets (Yin & Neubig, 2022), avoids using reference translations to stay clear of problematic distributional shifts (Vamvas & Sennrich, 2021b), and can be applied on unannotated inputs to identify context usage in model generations.

After formalizing our proposed approach in Section 3, we apply PECoRe to contextual machine translation (MT) to study the plausibility of context reliance in bilingual and multilingual MT models. While PECoRe can easily be used alongside encoder-decoder and decoder-only language models for interpreting context usage in any text generation task, we focus our evaluation on MT because of its constrained output space facilitating automatic assessment and the availability of MT datasets annotated with human rationales of context usage. We thoroughly test PECoRe on well-known discourse phenomena, benchmarking several context sensitivity metrics and attribution methods to identify cue-target pairs. We conclude by applying PECoRe to unannotated examples and showcasing some reasonable and questionable cases of context reliance in MT model translations.

In sum, we make the following contributions[1]:

- We introduce PECoRe, an interpretability framework to detect and attribute context reliance in language models. PECoRe enables a quantitative evaluation of plausibility for language generation beyond the limited artificial settings explored in previous literature.

- We compare the effectiveness of context sensitivity metrics and feature attribution methods on the context-aware MT tasks, showing the limitations of metrics currently in use.

- We apply PECoRe to naturally-occurring translations to identify interesting discourse-level phenomena and discuss issues in the context usage abilities of context-aware MT models.

## 2 RELATED WORK

**Context Usage in Language Generation** An appropriate[2] usage of input information is fundamental in tasks such as summarization (Maynez et al., 2020) to ensure the soundness of generated texts. While appropriateness is traditionally verified post-hoc using trained models (Durmus et al., 2020; Kryscinski et al., 2020; Goyal & Durrett, 2021), recent interpretability works aim to gauge input influence on model predictions using internal properties of language models, such as the mixing of contextual information across model layers (Kobayashi et al., 2020; Ferrando et al., 2022b; Mohebbi et al., 2023) or the layer-by-layer refinement of next token predictions (Geva et al., 2022; Belrose et al., 2023). Recent attribution methods can disentangle factors influencing generation in language models (Yin & Neubig, 2022) and were successfully used to detect and mitigate hallucinatory behaviors (Tang et al., 2022; Dale et al., 2022; 2023). Our proposed method adopts this intrinsic perspective to identify context reliance without ad-hoc trained components.

---

[1]Code: https://github.com/gsarti/pecore. The CLI command inseq attribute-context available in the Inseq library is a generalized PECoRe implementation: https://github.com/inseq-team/inseq

[2]We avoid using the term *faithfulness* due to its ambiguous usage in interpretability research.

**Context Usage in Neural Machine Translation**    Inter-sentential context is often fundamental for resolving discourse-level ambiguities during translation (Müller et al., 2018; Bawden et al., 2018; Voita et al., 2019b; Fernandes et al., 2023). However, MT systems are generally trained at the sentence level and fare poorly in realistic translation settings (Läubli et al., 2018; Toral et al., 2018). Despite advances in context-aware MT (Voita et al., 2018; 2019a; Lopes et al., 2020; Majumder et al., 2022; Jin et al., 2023 *inter alia*, surveyed by Maruf et al., 2021), only a few works explored whether context usage in MT models aligns with human intuition. Notably, some studies focused on *which parts of context* inform model predictions, finding that supposedly context-aware MT models are often incapable of using contextual information (Kim et al., 2019; Fernandes et al., 2021) and tend to pay attention to irrelevant words (Voita et al., 2018), with an overall poor agreement between human annotations and model rationales (Yin et al., 2021). Other works instead investigated *which parts of generated texts* are influenced by context, proposing various contrastive methods to detect gender biases, over/under-translations (Vamvas & Sennrich, 2021a; 2022), and to identify various discourse-level phenomena in MT corpora (Fernandes et al., 2023). While these two directions have generally been investigated separately, our work proposes a unified framework to enable an end-to-end evaluation of context-reliance plausibility in language models.

**Plausibility of Model Rationales**    Plausibility evaluation for NLP models has largely focused on classification models (DeYoung et al., 2020; Atanasova et al., 2020; Attanasio et al., 2023). While few works investigate plausibility in language generation (Vafa et al., 2021; Ferrando et al., 2023), such evaluations typically involve a single generation step to complete a target sentence with a token connected to preceding information (e.g. subject/verb agreement, as in "*The pictures of the cat [is/are]*"), effectively biasing the evaluation by using a pre-selected token of interest. On the contrary, our framework proposes a more comprehensive evaluation of generation plausibility that includes the identification of context-sensitive generated tokens as an important prerequisite. Additional background on rationales and plausibility evaluation is provided in Appendix A.

## 3   THE PECoRe FRAMEWORK

PECoRe is a two-step framework for identifying context dependence in generative language models. First, *context-sensitive tokens identification* (CTI) selects which tokens among those generated by the model were influenced by the presence of the preceding context (e.g. the feminine options "`alla pastorella, le`" in Figure 1). Then, *contextual cues imputation* (CCI) attributes the prediction of context-sensitive tokens to specific cues in the provided context (e.g. the feminine cues "`she, Le pecore`" in Figure 1). **Cue-target pairs** formed by influenced target tokens and their respective influential context cues can then be compared to human rationales to assess the models' plausibility of context reliance for contextual phenomena of interest. Figure 2 provides an overview of the two steps applied to the context-aware MT setting discussed by this work. A more general formalization of the framework for language generation is proposed in the following sections.

**Notation**    Let $X_{\text{ctx}}^i$ be the sequence of contextual inputs containing $N$ tokens from vocabulary $\mathcal{V}$, composed by current input $x$, generation prefix $y_{<i}$ and context $C$. Let $X_{\text{no-ctx}}^i$ be the non-contextual input in which $C$ tokens are excluded.[3] $P_{\text{ctx}}^i = P(x, y_{<i}, C, \theta)$ is the discrete probability distribution over $\mathcal{V}$ at generation step $i$ of a language model with $\theta$ parameters receiving contextual inputs $X_{\text{ctx}}^i$. Similarly, $P_{\text{no-ctx}}^i = P(x, y_{<i}, \theta)$ is the distribution obtained from the same model for non-contextual input $X_{\text{no-ctx}}^i$. Both distributions are equivalent to vectors in the probability simplex in $\mathbb{R}^{|\mathcal{V}|}$, and we use $P_{\text{ctx}}(y_i)$ to denote the probability of next token $y_i$ in $P_{\text{ctx}}^i$, i.e. $P(y_i \mid x, y_{<i}, C)$.

### 3.1   CONTEXT-SENSITIVE TOKEN IDENTIFICATION

CTI adapts the contrastive conditioning paradigm by Vamvas & Sennrich (2021a) to detect input context influence on model predictions using the contrastive pair $P_{\text{ctx}}^i, P_{\text{no-ctx}}^i$. Both distributions are relative to the **contextual target sentence** $\hat{y} = \{\hat{y}_1 \ldots \hat{y}_n\}$, corresponding to the sequence produced by a decoding strategy of choice in the presence of input context. In Figure 2, the contextual target sentence $\hat{y} = $ "*Sont-elles à l'hôtel?*" is generated when $x$ and contexts $C_x, C_{\hat{y}}$ are provided as inputs, while **non-contextual target sentence** $\tilde{y} = $ "*Ils sont à l'hôtel?*" would be produced when only $x$ is

---

[3]In the context-aware MT example of Figure 2, $C$ includes source context $C_x$ and target context $C_y$.

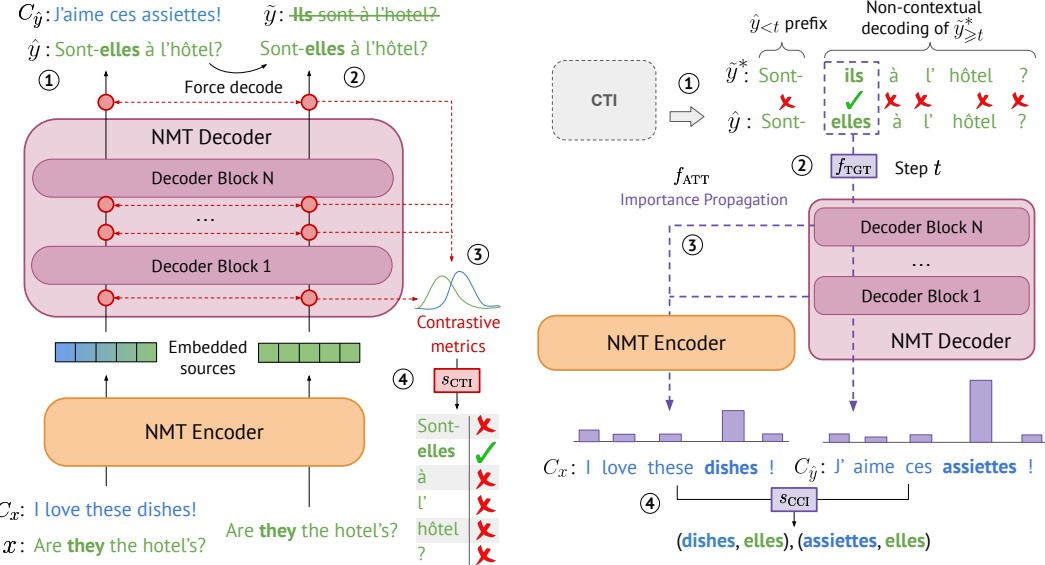

Figure 2: The PECoRe framework. **Left:** Context-sensitive token identification (CTI). ①: A context-aware MT model translates source context ($C_x$) and current ($x$) sentences into target context ($C_{\hat{y}}$) and current ($\hat{y}$) outputs. ②: $\hat{y}$ is force-decoded in the non-contextual setting instead of natural output $\tilde{y}$. ③: Contrastive metrics are collected throughout the model for every $\hat{y}$ token to compare the two settings. ④: Selector $s_{\text{CTI}}$ maps metrics to binary context-sensitive labels for every $\hat{y}_i$. **Right:** Contextual cues imputation (CCI). ①: Non-contextual target $\tilde{y}^*$ is generated from contextual prefix $\hat{y}_{<t}$. ②: Function $f_{\text{TGT}}$ is selected to contrast model predictions with ($\hat{y}_t$) and without ($\tilde{y}_t^*$) input context. ③: Attribution method $f_{\text{ATT}}$ using $f_{\text{TGT}}$ as target scores contextual cues driving $\hat{y}_t$ prediction. ④: Selector $s_{\text{CCI}}$ selects relevant cues, and cue-target pairs are assembled.

provided. In the latter case, $\hat{y}$ is instead force-decoded from the non-contextual setting to enable a direct comparison of matching outputs. We define a set of **contrastive metrics** $\mathcal{M} = \{m_1, \ldots, m_M\}$, where each $m : \Delta_{|\mathcal{V}|} \times \Delta_{|\mathcal{V}|} \mapsto \mathbb{R}$ maps a contrastive pair of probability vectors to a continuous score. For example, the difference in next token probabilities for contextual and non-contextual settings, i.e. $P_{\text{diff}}(\hat{y}_i) = P_{\text{ctx}}(\hat{y}_i) - P_{\text{no-ctx}}(\hat{y}_i)$, might be used for this purpose.[4] Target tokens with high contrastive metric scores can be identified as *context-sensitive*, provided $C$ is the only added parameter in the contextual setting. Finally, a **selector** function $s_{\text{CTI}} : \mathbb{R}^{|\mathcal{M}|} \mapsto \{0, 1\}$ (e.g. a statistical threshold selecting salient scores) is used to classify every $\hat{y}_i$ as context-sensitive or not.

## 3.2 CONTEXTUAL CUES IMPUTATION

CCI applies the contrastive attribution paradigm (Yin & Neubig, 2022) to trace the generation of every context-sensitive token in $\hat{y}$ back to context $C$, identifying cues driving model predictions.

**Definition 3.1.** Let $\mathcal{T}$ be the set of indices corresponding to context-sensitive tokens identified by the CTI step, such that $t \in \hat{y}$ and $\forall t \in \mathcal{T}, s_{\text{CTI}}(m_1^t, \ldots, m_M^t) = 1$. Let also $f_{\text{TGT}} : \Delta_{|\mathcal{V}|} \times \Delta_{|\mathcal{V}|} \mapsto \mathbb{R}$ be a **contrastive attribution target** function having the same domain and range as metrics in $\mathcal{M}$. The **contrastive attribution method** $f_{\text{ATT}}$ is a composite function quantifying the importance of contextual inputs to determine the output of $f_{\text{TGT}}$ for a given model with $\theta$ parameters.

$$f_{\text{ATT}}(\hat{y}_t) = f_{\text{ATT}}(x, \hat{y}_{<t}, C, \theta, f_{\text{TGT}}) = f_{\text{ATT}}\big(x, \hat{y}_{<t}, C, \theta, f_{\text{TGT}}(P_{\text{ctx}}^t, P_{\text{no-ctx}}^t)\big) \tag{1}$$

**Remark 3.1.** Distribution $P_{\text{no-ctx}}^t$ in Equation (1) is from the contextual prefix $\hat{y}_{<t} = \{\hat{y}_1, \ldots, \hat{y}_{t-1}\}$ (e.g. $\hat{y}_{<t}$ ="*Sont-*" in Figure 2) and non-contextual inputs $X_{\text{no-ctx}}^t$. This is conceptually equivalent to predicting the next token of a new non-contextual sequence $\tilde{y}^*$ which, contrary to $\tilde{y}$, starts from a forced contextual prefix $\hat{y}_{<t}$ (e.g. "*ils*" in $\tilde{y}^*$ = "*Sont-ils à l'hôtel?*" in Figure 2).

**Remark 3.2.** Provided that $P_{\text{ctx}}^t$ and $P_{\text{no-ctx}}^t$ depend respectively on contextual and non-contextual inputs $X_{\text{ctx}}^t, X_{\text{no-ctx}}^t$ despite using the same prefix $\hat{y}_{<t}$, probabilities $P_{\text{ctx}}(\hat{y}_t), P_{\text{no-ctx}}(\tilde{y}_t^*)$ are likely to

---

[4]We use $m^i$ to denote the result of $m\big(P_{\text{ctx}}^i, P_{\text{no-ctx}}^i\big)$. Several metrics are presented in Section 4.2.

differ even when $\hat{y}_t = \tilde{y}_t^*$, i.e. even when the next predicted token is the same, it is likely to have a different probability in the two settings, ultimately resulting in non-zero $f_{\text{TGT}}$ and $f_{\text{ATT}}(\hat{y}_t)$ scores.

**Remark 3.3.** Our formalization of $f_{\text{ATT}}$ generalizes the method proposed by (Yin & Neubig, 2022) to support any target-dependent attribution method, such as popular gradient-based approaches (Simonyan et al., 2014; Sundararajan et al., 2017), and any contrastive attribution target $f_{\text{TGT}}$.[5]

$f_{\text{ATT}}$ produces a sequence of attribution scores $A_t = \{a_1, \ldots, a_N\}$ matching contextual input length $N$. From those, only the subset $A_{t\,\text{CTX}}$ of scores corresponding to context input sequence $C$ are passed to **selector** function $s_{\text{CCI}} : \mathbb{R} \mapsto \{0, 1\}$, which predicts a set $\mathcal{C}_t$ of indices corresponding to contextual cues identified by CCI, such that $\forall c \in \mathcal{C}_t, \forall a \in A_{t\,\text{CTX}}, s_{\text{CCI}}(a_c) = 1$.

Having collected all context-sensitive generated token indices $\mathcal{T}$ using CTI and their contextual cues through CCI ($C_t$), PECoRe ultimately returns a sequence $S_{\text{ct}}$ of all identified cue-target pairs:

$$\mathcal{T} = \text{CTI}(C, x, \hat{y}, \theta, \mathcal{M}, s_{\text{CTI}}) = \{t \mid s_{\text{CTI}}(m_1^t, \ldots, m_M^t) = 1\}$$
$$\mathcal{C} = \text{CCI}(\mathcal{T}, C, x, \hat{y}, \theta, f_{\text{ATT}}, f_{\text{TGT}}, s_{\text{CCI}}) = \{c \mid s_{\text{CCI}}(a_c) = 1 \,\forall a_c \in A_{t\,\text{ctx}}, \forall t \in \mathcal{T}\} \quad (2)$$
$$S = \text{PECoRe}(C, x, \theta, s_{\text{CTI}}, s_{\text{CCI}}, \mathcal{M}, f_{\text{ATT}}, f_{\text{TGT}}) = \{(C_c, \hat{y}_t) \mid \forall t \in \mathcal{T}, \forall c \in \mathcal{C}_t, \forall \mathcal{C}_t \in \mathcal{C}\}$$

## 4 Context Reliance Plausibility in Context-aware MT

This section describes our evaluation of PECoRe in a controlled setup. We experiment with several contrastive metrics and attribution methods for CTI and CCI (Section 4.2, Section 4.4), evaluating them in isolation to quantify the performance of individual components. An end-to-end evaluation is also performed in Section 4.4 to establish the applicability of PECoRe in a naturalistic setting.

### 4.1 Experimental Setup

**Evaluation Datasets**   Evaluating generation plausibility requires human annotations for context-sensitive tokens in target sentences and disambiguating cues in their preceding context. To our knowledge, the only resource matching these requirements is SCAT Yin et al. (2021), an English→French corpus with human annotations of anaphoric pronouns and disambiguating context on OpenSubtitles2018 dialogue translations (Lison et al., 2018; Lopes et al., 2020). SCAT examples were extracted automatically using lexical heuristics and thus contain only a limited set of anaphoric pronouns (*it, they → il/elle, ils/elles*), with no guarantees of contextual cues being found in preceding context. To improve our assessment, we select a subset of high-quality SCAT test examples containing contextual dependence, which we name SCAT+. Additionally, we manually annotate contextual cues in DiscEval-MT (Bawden et al., 2018), another English→French corpus containing handcrafted examples for *anaphora resolution* (ANA) and *lexical choice* (LEX). Our final evaluation set contains 250 SCAT+ and 400 DiscEval-MT translations across two discourse phenomena.[6]

**Models**   We evaluate two bilingual OpusMT models (Tiedemann & Thottingal, 2020) using the Transformer base architecture (Vaswani et al., 2017) (Small and Large), and mBART-50 1-to-many (Tang et al., 2021), a larger multilingual MT model supporting 50 target languages, using the Transformers library (Wolf et al., 2020). We fine-tune models using extended translation units (Tiedemann & Scherrer, 2017) with contextual inputs marked by break tags such as "`source context <brk> source current`" to produce translations in the format "`target context <brk> target current`", where context and current target sentences are generated[7]. We perform context-aware fine-tuning on 242k IWSLT 2017 English→French examples (Cettolo et al., 2017), using a dynamic context size of 0-4 preceding sentences to ensure robustness to different context lengths and allow contextless usage. To further improve models' context sensitivity, we continue fine-tuning on the SCAT training split, containing 11k examples with inter- and intra-sentential pronoun anaphora.

**Model Disambiguation Accuracy**   We estimate contextual disambiguation accuracy by verifying whether annotated (gold) context-sensitive words are found in model outputs. Results before and after

---

[5]Additional precisions and formalization of target-dependent attribution methods are provided in Appendix B.

[6]SCAT+: `https://hf.co/datasets/inseq/scat`. DiscEval-MT: `https://hf.co/datasets/inseq/disc_eval_mt`. Appendix E describes the annotation process and presents some examples for the two datasets.

[7]Context-aware MT model using only source context are also evaluated in Section 4.5 and Appendix D

| | SCAT+ | | | DISCEVAL-MT (ANA) | | | DISCEVAL-MT (LEX) | | |
|---|---|---|---|---|---|---|---|---|---|
| **Model** | **BLEU** | **OK** | **OK-CS** | **BLEU** | **OK** | **OK-CS** | **BLEU** | **OK** | **OK-CS** |
| OpusMT Small *(default)* | 29.1 | 0.14 | - | 43.9 | 0.40 | - | 30.5 | 0.29 | - |
| OpusMT Small S+T$_{ctx}$ | 39.1 | 0.81 | 0.59 | 48.1 | 0.60 | 0.24 | 33.5 | 0.36 | 0.07 |
| OpusMT Large *(default)* | 29.0 | 0.16 | - | 39.2 | 0.41 | - | 31.2 | 0.31 | - |
| OpusMT Large S+T$_{ctx}$ | **40.3** | **0.83** | 0.58 | 48.9 | **0.68** | 0.31 | **34.8** | **0.38** | 0.10 |
| mBART-50 *(default)* | 23.8 | 0.26 | - | 33.4 | 0.42 | - | 24.5 | 0.25 | - |
| mBART-50 S+T$_{ctx}$ | 37.6 | 0.82 | 0.55 | **49.0** | 0.62 | 0.32 | 29.3 | 0.30 | 0.07 |

Table 1: Translation quality of EN → FR MT models before *(default)* and after *(S+T$_{ctx}$)* context-aware MT fine-tuning. **OK**: % of translations with correct disambiguation for discourse phenomena. **OK-CS**: % of translations where the correct disambiguation is achieved only when context is provided.

context-aware fine-tuning are shown in Table 1. We find that fine-tuning improves translation quality and disambiguation accuracy across all tested models, with larger gains for anaphora resolution datasets closely matching fine-tuning data. To gain further insight into these results, we use context-aware models to translate examples with and without context and identify a subset of *context-sensitive translations* (OK-CS) for which the correct target word is generated only when input context is provided to the model. Interestingly, we find a non-negligible amount of translations that are correctly disambiguated even in the absence of input context (corresponding to OK minus OK-CS in Table 1). For these examples, the correct prediction of ambiguous words aligns with model biases, such as defaulting to masculine gender for anaphoric pronouns (Stanovsky et al., 2019) or using the most frequent sense for word sense disambiguation. Provided that such examples are unlikely to exhibit context reliance, we focus particularly on the OK-CS subset results in our following evaluation.

## 4.2 METRICS FOR CONTEXT-SENSITIVE TARGET IDENTIFICATION

The following contrastive metrics are evaluated for detecting context-sensitive tokens in the CTI step.

**Relative Context Saliency** We use contrastive gradient norm attribution (Yin & Neubig, 2022) to compute input importance towards predicting the next token $\hat{y}_i$ with and without input context. Positive importance scores are obtained for every input token using the L2 gradient vectors norm (Bastings et al., 2022), and relative context saliency is obtained as the proportion between the normalized importance for context tokens $c \in C_x, C_y$ and the overall input importance, following previous work quantifying MT input contributions (Voita et al., 2021; Ferrando et al., 2022a; Edman et al., 2023).

$$\nabla_{\text{ctx}}(P_{\text{ctx}}^i, P_{\text{no-ctx}}^i) = \frac{\sum_{c \in C_x, C_y} \left\| \nabla_c \left( P_{\text{ctx}}(\hat{y}_i) - P_{\text{no-ctx}}(\hat{y}_i) \right) \right\|}{\sum_{t \in X_{\text{ctx}}^i} \left\| \nabla_t \left( P_{\text{ctx}}(\hat{y}_i) - P_{\text{no-ctx}}(\hat{y}_i) \right) \right\|} \tag{3}$$

**Likelihood Ratio (LR)** and **Pointwise Contextual Cross-mutual Information (P-CXMI)** Proposed by Vamvas & Sennrich (2021a) and Fernandes et al. (2023) respectively, both metrics frame context dependence as a ratio of contextual and non-contextual probabilities.

$$\text{LR}(P_{\text{ctx}}^i, P_{\text{no-ctx}}^i) = \frac{P_{\text{ctx}}(\hat{y}_i)}{P_{\text{ctx}}(\hat{y}_i) + P_{\text{no-ctx}}(\hat{y}_i)} \quad (4) \quad \text{P-CXMI}(P_{\text{ctx}}^i, P_{\text{no-ctx}}^i) = -\log \frac{P_{\text{ctx}}(\hat{y}_i)}{P_{\text{no-ctx}}(\hat{y}_i)} \quad (5)$$

**KL-Divergence** (Kullback & Leibler, 1951) between $P_{\text{ctx}}^i$ and $P_{\text{no-ctx}}^i$ is the only metric we evaluate that considers the full distribution rather than the probability of the predicted token. We include it to test the intuition that the impact of context inclusion might extend beyond top-1 token probabilities.

$$D_{\text{KL}}(P_{\text{ctx}}^i \| P_{\text{no-ctx}}^i) = \sum_{\hat{y}_i \in \mathcal{V}} P_{\text{ctx}}(\hat{y}_i) \log \frac{P_{\text{ctx}}(\hat{y}_i)}{P_{\text{no-ctx}}(\hat{y}_i)} \tag{6}$$

## 4.3 CTI PLAUSIBILITY RESULTS

Figure 3 presents our metrics evaluation for CTI, with results for the full test sets and the subsets of context-sensitive sentences (OK-CS) highlighted in Table 1. To keep our evaluation simple, we use a naive $s_{\text{CTI}}$ selector tagging all tokens with metric scores one standard deviation above the per-example mean as context-sensitive. We also include a stratified random baseline matching the frequency of

Figure 3: Macro F1 of contrastive metrics for context-sensitive target token identification (CTI) using OpusMT Large on the full datasets (left) or on OK-CS context-sensitive subsets (right).

occurrence of context-sensitive tokens in each dataset. Datapoints in Figure 3 are sentence-level macro F1 scores computed for every dataset example. Full results are available in Appendix G.

Pointwise metrics (LR, P-CXMI) show high plausibility for the context-sensitive subsets OK-CS across all datasets and models but achieve lower performances on the full test set, especially for lexical choice phenomena less present in MT models' training. KL-Divergence performs on par or better than pointwise metrics, suggesting that distributional shifts beyond top prediction candidates can provide useful information to detect context sensitivity. On the contrary, the poor performance of context saliency indicates that context reliance in aggregate cannot reliably predict context sensitivity. A manual examination of misclassified examples reveals several context-sensitive tokens that were not annotated as such since they did not match datasets' phenomena of interest but were still identified by CTI metrics (examples in Appendix G). This further underscores the importance of data-driven end-to-end approaches like PECORE to limit the influence of selection bias during evaluation.

## 4.4 METHODS FOR CONTEXTUAL CUES IMPUTATION

The following attribution methods are evaluated for detecting contextual cues in the CCI step.

**Contrastive Gradient Norm** (Yin & Neubig, 2022) estimates input tokens' contributions towards predicting a target token instead of a contrastive alternative. We use this method to explain the generation of context-sensitive tokens in the presence and absence of context.

$$A_{t\,\mathrm{ctx}} = \{\, \|\nabla_c\big(f_{\mathrm{TGT}}(P^i_{\mathrm{ctx}}, P^i_{\mathrm{no\text{-}ctx}})\big)\| \,|\, \forall c \in C\} \tag{7}$$

For the choice of $f_{\mathrm{tgt}}$, we evaluate both probability difference $P_{\mathrm{ctx}}(\hat{y}_i) - P_{\mathrm{no\text{-}ctx}}(\hat{y}_i)$, conceptually similar to the original formulation, and the KL-Divergence of contextual and non-contextual distributions $D_{\mathrm{KL}}(P^i_{\mathrm{ctx}}\|P^i_{\mathrm{no\text{-}ctx}})$. We use $\nabla_{\mathrm{diff}}$ and $\nabla_{\mathrm{KL}}$ to identify gradient norm attribution in the two settings. $\nabla_{\mathrm{KL}}$ scores can be seen as the contribution of input tokens towards the shift in probability distribution caused by the presence of input context.

**Attention Weights** Following previous work, we use the mean attention weight across all heads and layers (Attention Mean, Kim et al., 2019) and the weight for the head obtaining the highest plausibility per-dataset (Attention Best, Yin et al., 2021) as importance measures for CCI. Attention Best can be seen as a best-case estimate of attention performance but is not a viable metric in real settings, provided that the best attention head to capture a phenomenon of interest is unknown beforehand. Since attention weights are model byproducts unaffected by predicted outputs, we use only attention scores for the contextual setting $P^i_{\mathrm{ctx}}$ and ignore the contextless alternative when using these metrics.

## 4.5 CCI PLAUSIBILITY RESULTS

We conduct a controlled CCI evaluation using gold context-sensitive tokens as a starting point to attribute contextual cues.[8] This corresponds to the baseline plausibility evaluation described in Section 2, allowing us to evaluate attribution methods in isolation, assuming perfect identification of context-sensitive tokens. Figure 4 presents our results. Scores in the right plot are relative to the context-aware OpusMT Large model of Section 4.3 using both source and target context. Instead, the

---

[8]To avoid using references as model generations, we align annotations to natural model outputs (Appendix F).

Figure 4: Macro F1 of CCI methods over full datasets using OpusMT Large models trained with only source context (left) or with source+target context (right). Boxes and red median lines show CCI results based on gold context-sensitive tokens. Dotted bars show median CCI scores obtained from context-sensitive tokens identified by KL-Divergence during CTI (E2E settings).

left plot presents results for an alternative version of the same model that was fine-tuned using only source context (i.e. translating $C_x, x \to y$ without producing target context $C_y$). Source-only context was used in previous context-aware MT studies (Fernandes et al., 2022), and we include it in our analysis to assess how the presence of target context impacts model plausibility. We finally validate the end-to-end plausibility of PECORE-detected pairs using context-sensitive tokens identified by the best CTI metric from Section 4.3 (KL-Divergence) as the starting point for CCI, and using a simple statistical selector equivalent to the one used for CTI evaluation. Results for the OK-CS subset are omitted as they show comparable trends. Full results are available in Appendix H.

First, contextual cues are more easily detected for the source-only model using all evaluated methods. This finding corroborates previous evidence highlighting how context usage issues might emerge when lengthy context is provided (Fernandes et al., 2021; Shi et al., 2023). When moving from gold CTI tags to the end-to-end setting (E2E) we observe a larger drop in plausibility for the SCAT+ and DISCEVAL-MT ANA datasets that more closely match the fine-tuning data of analyzed MT models. This suggests that standard evaluation practices may overestimate model plausibility for in-domain settings and that our proposed framework can effectively mitigate this issue. Interestingly, the Attention Best method suffers the most from end-to-end CCI application, while other approaches are more mildly affected. This can result from attention heads failing to generalize to other discourse-level phenomena at test time, providing further evidence of the limitations of attention as an explanatory metric (Jain & Wallace, 2019; Bastings & Filippova, 2020). While $\nabla_{\text{KL}}$ and $\nabla_{\text{diff}}$ appear as the most robust choice across the two datasets, per-example variability remains high across the board, leaving space for improvement for more plausible attribution methods in future work.

## 5 DETECTING CONTEXT RELIANCE IN THE WILD

We conclude our analysis by applying the PECORE method to the popular Flores-101 MT benchmark (Goyal et al., 2022), containing groups of 3-5 contiguous sentences from English Wikipedia. While in previous sections, labeled examples were used to evaluate the effectiveness of PECORE components, here we apply our framework end-to-end to unannotated MT outputs and inspect resulting cue-target pairs to identify successes and failures of context-aware MT models. Specifically, we apply PECORE to the context-aware OpusMT Large and mBART-50 models of Section 4.1, using KL-Divergence as CTI metric and $\nabla_{\text{KL}}$ as CCI attribution method. We set $s_{\text{CTI}}$ and $s_{\text{CCI}}$ to two standard deviations above the per-example average score to focus our analysis on very salient tokens.

Table 2 shows some examples annotated with PECORE outputs, with more examples available in Appendix I. In the first example, the acronym MS, standing for Multiple Sclerosis, is translated generically as *la maladie* (the illness) in the contextual output, but as *SEP* (the French acronym for MS, i.e. *sclérose en plaques*) when context is not provided. PECORE shows how this choice is mostly driven by the MS mention in source context $C_x$ while the term *sclérose en plaques* in target context $C_y$ is not identified as influential, possibly motivating the choice for the more generic option.

In the second example, the prediction of pronoun *elles* (they, feminine) depends on the context noun phrase *mob of market women* (*foule de femmes du marché* in French). However, the correct pronoun referent is *Le roi et Madame Elizabeth* (*the king and Madam Elizabeth*), so the pronoun should be

---

**1. Acronym Translation (English → French, correct but more generic)**

$C_x$ : Across the United States of America, there are approximately 400,000 known cases of Multiple Sclerosis (MS) [...]
$C_y$ : Aux États-Unis, il y a environ 400 000 cas connus de sclérose en plaques [...]
$x$ : MS affects the central nervous system, which is made up of the brain, the spinal cord and the optic nerve.
$\tilde{y}$ : La SEP affecte le système nerveux central, composé du cerveau, de la moelle épinière et du nerf optique.
$\hat{y}$ : La maladie affecte le système nerveux central, composé du cerveau, de la moelle épinière et du nerf optique.

**2. Anaphora Resolution (English → French, incorrect)**

$C_x$ : The terrified King and Madam Elizabeth were forced back to Paris by a mob of market women.
$C_y$ : Le roi et Madame Elizabeth ont été forcés à revenir à Paris par une foule de femmes du marché.
$x$ : In a carriage, they traveled back to Paris surrounded by a mob of people screaming and shouting threats [...]
$\tilde{y}$ : Dans une carriole, ils sont retournés à Paris entourés d'une foule de gens hurlant et criant des menaces [...]
$\hat{y}$ : Dans une carriole, elles sont *retournées* à Paris *entourées* d'une foule de gens hurlant et criant des menaces [...]

**3. Numeric format cohesion (English → French, incorrect)**

$C_x$ : The games kicked off at 10:00am with great weather apart from mid morning drizzle [...]
$C_y$ : Les matchs se sont écoulés à 10:00 du matin avec un beau temps à part la nuée du matin [...]
$x$ : South Africa started on the right note when they had a comfortable 26-00 win against Zambia.
$\tilde{y}$ : L'Afrique du Sud a commencé sur la bonne note quand ils ont eu une confortable victoire de 26 contre le Zambia.
$\hat{y}$ : L'Afrique du Sud a commencé sur la bonne note quand ils ont eu une confortable victoire de 26:00 contre le Zambia.

**4. Lexical cohesion (English → Turkish, correct)**

$C_x$ : The activity of all stars in the system was found to be driven by their luminosity, their rotation, and nothing else.
$C_y$ : Sistemdeki bütün ulduzların faaliyetlerinin, parlaklıkları, rotasyonları ve başka hiçbir şeyin etkisi altında olduğunu ortaya çıkardılar.
$x$ : The luminosity and rotation are used together to determine a star's Rossby number, which is related to plasma flow.
$\tilde{y}$ : Parlaklık ve döngü, bir *yıldızın plazm* akışıyla ilgili Rossby sayısını belirlemek için birlikte kullanılıyor.
$\hat{y}$ : Parlaklık ve rotasyon, bir *ulduzun plazma* akışıyla ilgili Rossby sayısını belirlemek için birlikte kullanılıyor.

---

Table 2: Flores-101 examples with cue-target pairs identified by PECoRe in OpusMT Large (1,2) and mBART-50 (3,4) contextual translations. Context-sensitive tokens generated instead of their non-contextual counterparts are identified by CTI, and contextual cues justifying their predictions are retrieved by CCI. *Other changes* in $\hat{y}$ are not considered context-sensitive by PECoRe.

the masculine default *ils* commonly used for mixed gender groups in French. PECoRe identifies this as a context-dependent failure due to an issue with the MT model's anaphora resolution. The third example presents an interesting case of erroneous numeric format cohesion that not be detected from pre-defined linguistic hypotheses. In this sentence, the score *26-00* is translated as *26* in the contextless output and as *26:00* in the context-aware translation. The *10:00* time indications found by PECoRe in the contexts suggest this is a case of problematic lexical cohesion.

Finally, we include an example of context usage for English→Turkish translation to test the contextual capabilities of the default mBART-50 model without context-aware fine-tuning. Again, PECoRe shows how the word *rotasyon* (rotation) is selected over *döngü* (loop) as the correct translation in the contextual case due to the presence of the lexically similar word *rotasyonları* in the previous context.

## 6 CONCLUSION

In this work, we introduced PECoRe, a novel interpretability framework to detect and attribute context usage in language models' generations. PECoRe extends the common plausibility evaluation procedure adopted in interpretability research by proposing a two-step procedure to identify context-sensitive generated tokens and match them to contextual cues contributing to their prediction. We applied PECoRe to context-aware MT, finding that context-sensitive tokens and their disambiguating rationales can be detected consistently and with reasonable accuracy across several datasets, models and discourse phenomena. Moreover, an end-to-end application of our framework without human annotations revealed incorrect context usage, leading to problematic MT model outputs.

While our evaluation is focused on the machine translation domain, PECoRe can easily be applied to other context-dependent language generation tasks such as question answering and summarization. Future applications of our methodology could investigate the usage of in-context demonstrations and chain-of-thought reasoning in large language models (Brown et al., 2020; Wei et al., 2022), explore PECoRe usage for different model architectures and input modalities (e.g. Appendix J), and pave the way for trustworthy citations in retrieval-augmented generation systems (Borgeaud et al., 2022).

ACKNOWLEDGMENTS

Gabriele Sarti, Grzegorz Chrupała and Arianna Bisazza acknowledge the support of the Dutch Research Council (NWO) as part of the project InDeep (NWA.1292.19.399). We thank the Center for Information Technology of the University of Groningen for providing access to the Hábrók high performance computing cluster used in fine-tuning and evaluation experiments.

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

# A   Background: Plausibility of Model Rationales in NLP

## A.1   Definitions

In post-hoc interpretability, a **rationale** of model behavior is an example-specific measure of how input features influence model predictions. When using feature attribution methods to interpret language models, these rationales correspond to a set of scores reflecting the contribution of every token in the input sequence towards the prediction of the next generated word (Madsen et al., 2022).

**Plausibility**, also referred to as "human-interpretability" (Lage et al., 2019), is a measure of "how convincing the interpretation is to humans" (Jacovi & Goldberg, 2020). It is important to note that plausibility does not imply **faithfulness**, i.e. how accurately the rationale reflects the true reasoning process of the model (Wiegreffe & Pinter, 2019), since a good explanation of model behavior might not align with human intuition.

## A.2   Example of Canonical Plausibility Evaluation for Language Models

Consider the following sentence, adapted from the BLiMP corpus (Warstadt et al., 2020).

$$x = \text{A \textbf{report} about the Impressionists \underline{\textbf{has}}/\underline{have} won the writing competition.}$$

For the sentence to be grammatically correct, the verb *to have* must be correctly inflected as *has* to agree with the preceding noun *report*. Hence, to evaluate the plausibility of a language model for this example, the model is provided with the prefix $x' = $ "A report about the Impressionists". Then, attribution scores are computed for every input token towards the prediction of *has* as the next token. Finally, we verify whether these scores identify the token *report* as the most important to predict *has*.

We remark that the choice of the pair *report-has* in the canonical procedure described above is entirely based on grammatical correctness, and other potential pairs not matching these constraints are not considered (e.g. the usage of *report* to predict *writing*). This common procedure might also cause reasonable behaviors to be labeled as implausible. For example, the indefinite article *A* might be identified as the most important token to predict *has* since it is forcibly followed by a singular noun and can co-occur with *has* more frequently than *report* in the model's training data. These limitations in the standard hypothesis-driven approach to plausibility evaluation motivate our proposal for PECoRe as a data-driven alternative.

## A.3   Metrics for Plausibility Evaluation

In practice, the attribution procedure from the example above produces a sequence $I_m$ of length $|x'|$ containing continuous importance scores produced by the attribution method, and these are compared to a sequence $I_h$ of the same length containing binary values, where 1s correspond to the cues identified by human annotators (in the example above, only *report*), while the rest of the values are set to 0. In our experiments, we use two common plausibility metrics introduced by (DeYoung et al., 2020):

**Token-level Macro F1** is the harmonic mean of precision and recall at the token level, using $I_h$ as the ground truth and a discretized version of $I_m$ as the prediction. Macro-averaging is used to account for the sparsity of cues in $I_h$, and the discretization is performed by means of the selector functions $s_{\text{CTI}}, s_{\text{CCI}}$ introduced in Section 3. We use this metric in the main analysis as the discretization step will likely reflect a more realistic plausibility performance, as it matches more closely the annotation process used to derive $I_h$. We note that Macro F1 can be considered a lower bound for plausibility, as the results depend heavily on the choice of the selector used for discretization.

**Area Under Precision-Recall Curve (AUPRC)** is computed as the area under the curve obtained by varying a threshold over token importance scores and computing the precision and recall for resulting discretized $I_m$ predictions while keeping $I_h$ as the ground truth. Contrary to Macro F1, AUPRC is selector-independent and accounts for tokens' relative ranking and degree of importance. Consequently, it can be seen as an upper bound for plausibility, as if the optimal selector was used. Since this is not likely to be the case in practice but can still prove useful, we include AUPRC results in Figure 6 and Figure 8.

## B  PRECISIONS ON TARGET-DEPENDENT ATTRIBUTION METHODS

**Definition B.1.** Let $s, s'$ be the resulting scores of two attribution target functions $f_{\text{TGT}}, f'_{\text{TGT}}$. An attribution method $f_{\text{ATT}}$ is **target-dependent** if importance scores $A$ are computed in relation to the outcome of its attribution target function, i.e. whenever the following condition is verified.

$$f_{\text{ATT}}(x, y_{<t}, C, \theta, s) \neq f_{\text{ATT}}(x, y_{<t}, C, \theta, s') \ \ \forall s \neq s' \tag{8}$$

In practice, common gradient-based attribution approaches (Simonyan et al., 2014; Sundararajan et al., 2017) are target-dependent as they rely on the outcome predicted by the model (typically the logit or the probability of the predicted class) as differentiation target to backpropagate importance to model input features. Similarly, perturbation-based approaches (Zeiler & Fergus, 2014) use the variation in prediction probability for the predicted class when noise is added to some of the model inputs to quantify the importance of the noised features.

On the contrary, recent approaches relying solely on model internals to define input importance are generally target-insensitive. For example, attention weights used as model rationales, either in their raw form or after a rollout procedure to obtain a unified score (Abnar & Zuidema, 2020), are independent of the predicted outcome. Similarly, value zeroing scores (Mohebbi et al., 2023) reflect only the representational dissimilarity across model layers before and after zeroing value vectors, and as such do not explicitly account for model predictions.

## C  PECORE IMPLEMENTATION

Algorithm 1 provides a pseudocode implementation of the PECORE cue-target pair extraction process formalized in Section 3.

## D  FULL TRANSLATION PERFORMANCE

Table 3 presents the translation quality and accuracy across all tested models. We compute BLEU using the SACREBLEU library (Post, 2018) with default parameters `nrefs:1|case:mixed|eff:no|tok:13a|smooth:exp|version:2.3.1` and compute COMET scores using COMET-22 (Rei et al., 2022) (v2.0.2). The models fine-tuned with source and target context clearly outperform the ones trained with source only, both in terms of generic translation quality and context-sensitive disambiguation accuracy. This motivates our choice to focus primarily on those models for our main analysis. All models are available in the following Huggingface organization: `https://hf.co/context-mt`. The $S_{\text{ctx}}$ models correspond to those matching `context-mt/scat-<MODEL_TYPE>-ctx4-cwd1-en-fr`, while $S + T_{\text{ctx}}$ models have the `context-mt/scat-<MODEL_TYPE>-target-ctx4-cwd0-en-fr` identifier.

## E  DATASETS ANNOTATION PROCEDURE

**SCAT+**  The original SCAT test set by Yin et al. (2021) contains 1000 examples with automatically identified context-sensitive pronouns *it/they* (marked by `<p>...<\p>`) and human-annotated contextual cues aiding their disambiguation (marked by `<hon>...<\hoff>`). Of these, we find 38 examples containing malformed tags and several more examples where an unrelated word containing *it* or *they* was wrongly marked as context-sensitive (e.g. `the soccer ball h<p>it</p> your chest`). Moreover, due to the original extraction process adopted for SCAT, there is no guarantee that contextual cues will be contained in the preceding context as they could also appear in the same sentence, defeating

---

**Algorithm 1:** PECORE cue-target extraction process

---

**Input:** $C, x$ – Input context and current sequences
$\qquad \theta$ – Model parameters
$\qquad s_{\text{CTI}}, s_{\text{CCI}}$ – Selector functions
$\qquad \mathcal{M}$ – Contrastive metrics
$\qquad f_{\text{ATT}}$ – Contrastive attribution method
$\qquad f_{\text{TGT}}$ – Contrastive attribution target function
**Output:** Sequence $S_{\text{ct}}$ of cue-target token pairs

Generate sequence $\hat{y}$ from inputs $C, x$ using any decoding strategy ;

**Context-sensitive Target Identification (CTI):**
$\mathcal{T}$ – Empty set to store indices of context-sensitive target tokens of $\hat{y}$ ;
**for** $\hat{y}_i \in \hat{y}$ **do**
$\qquad$ **for** $m \in \mathcal{M}$ **do**
$\qquad\qquad m^i = m_j\big(P_{\text{ctx}}(\hat{y}_i), P_{\text{no-ctx}}(\hat{y}_i)\big)$ ;
$\qquad$ **if** $s_{\text{CTI}}(m_1^i, \ldots, m_M^i) = 1$ **then**
$\qquad\qquad$ Store $i$ in set $\mathcal{T}$ ;

**Contextual Cues Imputation (CCI):**
$S$ – Empty sequence to store cue-target token pairs ;
**for** $t \in \mathcal{T}$ **do**
$\qquad$ Generate constrained non-contextual target current sequence $\tilde{y}^*$ from $\hat{y}_{<t}$;
$\qquad$ Use attribution method $f_{\text{ATT}}$ using $f_{\text{TGT}}$ as attribution target to get input importance scores $A_t$;
$\qquad$ Identify the subset $A_{t\,\text{CTX}}$ corresponding to tokens of context $C = \{C_1, \ldots, C_K\}$ ;
$\qquad$ **for** $a_i \in A_{t\,\text{CTX}} = \{a_1, \ldots, a_K\}$ **do**
$\qquad\qquad$ **if** $s_{\text{CCI}}(a_i) = 1$ **then**
$\qquad\qquad\qquad$ Store $(C_i, \hat{y}_t)$ in $S_{\text{ct}}$
**return** $S$

---

| Model | SCAT+ | | | | DISCEVAL-MT (ANA) | | | | DISCEVAL-MT (LEX) | | | |
|---|---|---|---|---|---|---|---|---|---|---|---|---|
| | **BLEU** | **COMET** | **OK** | **OK-CS** | **BLEU** | **COMET** | **OK** | **OK-CS** | **BLEU** | **COMET** | **OK** | **OK-CS** |
| OpusMT Small *(default)* | 29.1 | .799 | 0.14 | - | 43.9 | .888 | 0.40 | - | 30.5 | .763 | 0.29 | - |
| OpusMT Small $S_{\text{ctx}}$ | 36.1 | .812 | 0.84 | 0.42 | 47.1 | **.900** | 0.61 | 0.28 | 28.3 | .764 | 0.31 | 0.05 |
| OpusMT Small S+T$_{\text{ctx}}$ | 39.1 | .816 | 0.81 | 0.59 | 48.1 | .889 | 0.60 | 0.24 | 33.5 | .774 | 0.36 | 0.07 |
| OpusMT Large *(default)* | 29.0 | .806 | 0.16 | - | 39.2 | .891 | 0.41 | - | 31.2 | .771 | 0.31 | - |
| OpusMT Large $S_{\text{ctx}}$ | 38.4 | .823 | 0.83 | 0.41 | 44.6 | .887 | 0.64 | 0.28 | 32.2 | .773 | 0.39 | 0.09 |
| OpusMT Large S+T$_{\text{ctx}}$ | **40.3** | **.827** | 0.83 | 0.58 | 48.9 | .896 | **0.68** | 0.31 | **34.8** | **.787** | 0.38 | 0.10 |
| mBART-50 *(default)* | 30.9 | .780 | 0.52 | - | 33.4 | .871 | 0.42 | - | 24.5 | .734 | 0.25 | - |
| mBART-50 $S_{\text{ctx}}$ | 33.5 | .808 | **0.87** | 0.42 | 36.3 | .869 | 0.57 | 0.23 | 25.7 | .760 | 0.29 | 0.06 |
| mBART-50 S+T$_{\text{ctx}}$ | 37.6 | .814 | 0.82 | 0.55 | **49.0** | .895 | 0.64 | 0.29 | 29.3 | .767 | 0.30 | 0.07 |

Table 3: Full model performances on EN $\rightarrow$ FR test sets before *(default)* and after context-aware MT fine-tuning. $S_{ctx}$ and S+T$_{ctx}$ are context-aware model variants using source-only and source+target context, respectively. **OK**: % of translations with correct disambiguation for discourse phenomena. **OK-CS**: % of translations where the correct disambiguation is achieved only when context is provided.

the purpose of our context usage evaluation. Thus, we prefilter the whole corpus to preserve only sentences with well-formed tags and inter-sentential contextual cues identified by original annotators. Moreover, a manual inspection procedure is carried out to validate the original cue tags and discard problematic sentences, obtaining a final set of 250 examples with inter-sentential pronoun coreference. SCAT+ is available on the Hugging Face Hub: `https://hf.co/datasets/inseq/scat`.

**DISCEVAL-MT** We use minimal pairs in the original dataset by Bawden et al. (2018) (e.g. the DISCEVAL-MT LEX examples in Table 4) to automatically mark differing tokens as context-sensitive. Then, contextual cues are manually labeled separately by two annotators with good familiarity with both English and French. Cue annotations are compared across the two splits, resulting in very high agreement due the simplicity of the corpus (97% overlap for ANA, 90% for LEX). The annotated version of DISCEVAL-MT is available on the Hugging Face Hub: `https://hf.co/datasets/inseq/disc_eval_mt`

| SCAT+ |
|---|
| $C_x$ : I loathe that song. But why did you bite poor Birdie's head off? Because I've heard it more times than I care to. It haunts me. Just stop, for a moment. 
 $C_y$ : Je hais cette chanson (song, FEMININE). Mais pourquoi avoir parlé ainsi à la pauvre Birdie ? Parce que j'ai entendu ce chant plus que de fois que je ne le peux. Elle (she) me hante. Arrêtez-vous un moment. 
 $x$ : How does it haunt you? 
 $y$ : Comment peut-elle (she) vous hanter? |
| $C_x$: - Ah! Sven! It's been so long. - Riley, it's good to see you. - You, too. How's the boat? Uh, it creaks, it groans. 
 $C_y$ : Sven ! - Riley, contente de te voir. - Content aussi. Comment va le bateau (boat, MASCULINE)? Il (he) craque de partout. 
 $x$ : Not as fast as it used to be. 
 $y$ : Il (he) n'est pas aussi rapide qu'avant. |
| DISCEVAL-MT ANA |
| $C_x$ : But how do you know the woman isn't going to turn out like all the others? 
 $C_y$ : Mais comment tu sais que la femme (woman, FEMININE) ne finira pas comme toutes les autres? 
 $x$ : This one's different. 
 $y$ : Celle-ci (This one, FEMININE) est différente. |
| $C_x$ : Can you authenticate these signatures, please? 
 $C_y$ : Pourriez-vous authentifier ces signatures (FEMININE), s'il vous plaît? 
 $x$ : Yes, they're mines. 
 $y$ : Oui, ce sont les miennes (mines, FEMININE). |
| DISCEVAL-MT LEX |
| $C_x$ : Do you think you can shoot it from here? 
 $C_y$ : Tu penses que tu peux le tirer (shoot) dessus à partir d'ici? 
 $x$ : Hand me that bow. 
 $y$ : Passe-moi cet arc (bow, WEAPON). |
| $C_x$ : Can I help you with the wrapping? 
 $C_y$ : Est-ce que je peux t'aider pour l'emballage (wrapping)? 
 $x$ : Hand me that bow. 
 $y$ : Passe-moi ce ruban (bow, GIFT WRAP). |

Table 4: Examples from the SCAT+ and DISCEVAL-MT datasets used in our analysis with highlighted context-sensitive tokens and contextual cues used for plausibility evaluation using PECORE. Glosses are added for French words of interest to facilitate understanding.

Table 4 presents some examples for the three splits. By design, SCAT+ sentences have more uniform context-sensitive targets (*it/they* → *il/elle/ils/elles*) and more naturalistic context with multiple cues to disambiguate the correct pronoun.

## F TECHNICAL DETAILS OF PECORE EVALUATION

**Aligning annotations** Provided that gold context-sensitive tokens are only available in annotated reference translations, a simple option when applying CCI to those would involve using references as model generations. However, this was shown to be problematic by previous research, as it would induce a *distributional discrepancy* in model predictions (Vamvas & Sennrich, 2021b). For this reason, we let the model generate a natural translation and instead try to align tags to this new sentence using the AWESOME aligner (Dou & Neubig, 2021) with LABSE multilingual embeddings (Feng et al., 2022). While this process is not guaranteed to always result in accurate tags, it provides a good approximation of gold CTI annotations on model generation for the purpose of our assessment.

## G FULL CTI RESULTS AND CTI PROBLEMATIC EXAMPLES

**CTI Results** Figure 5 and Figure 6 present the CTI plausibility of all tested models for the Macro F1 and AUPRC metrics, similarly to Figure 3 in the main analysis.

**CTI Problematic Examples** Table 5 shows some examples of OpusMT Large S+T$_{ctx}$ translations considered incorrect during CTI evaluation as the highlighted word do not match the gold SCAT+ labels. However, these words are correctly identified as context-sensitive by CTI metrics as they reflect the grammatical pronoun formality adopted in preceding French contexts $C_y$. The reason behind such mismatch is that SCAT+ annotations focus solely on gender disambiguation for anaphoric pronouns. Instead, CTI metrics detect all kinds of context dependence, including the formality cohesion shown

| Pronoun Grammatical Formality, SCAT+ |
|---|
| $C_x$ : Oh. Hi, Dr. Owens. My son posted on his Facebook page that he has a bullet in his lung. [...] 
 $C_y$ : Salut, Dr. Owens. Mon fils a posté sur sa page Facebook qu'il a une balle dans son poumon [...] 
 $x$ : And when the soccer ball hit your chest, it dislodged it. [...] 
 $y$ : Et quand la balle de football touche votre (your, 2ND P. PLUR., FORMAL) poitrine, elle la déplace. [...] |
| $C_x$ : [...] That demon that was in you, it wants you. But not like before. I think it loves you. 
 $C_y$ : [...] Ce démon qui était en vous, il vous veut. Mais pas comme avant. Je pense qu'il vous aime. 
 $x$ : And it's powerless without you. 
 $y$ : Et il est impuissant sans vous (you, 2ND P. PLUR., FORMAL). |
| $C_x$ : You threaten my father again, I'll kill you myself... on this road. You hear me? My quarrel was with your father. 
 $C_y$ : Tu menaces encore mon père, je te tuerai moi-même... sur cette route. Tu m'entends? Ma querelle était avec ton père. 
 $x$ : Now it is with you as well. 
 $y$ : Maintenant elle est aussi avec toi (you, 2ND P. SING., INFORMAL). |
| $C_x$ : She went back to Delhi. What do you think? [...] Girls, I tell you. 
 $C_y$ : Elle est revenue à Delhi. Qu'en penses-tu? [...] Les filles, je te le dis. 
 $x$ : I wish they were all like you. 
 $y$ : J'aimerais qu'elles soient toutes comme toi (you, 2ND P. SING., INFORMAL). |

Table 5: Examples of SCAT+ sentences with context-sensitive words identified by CTI but not originally labeled as context-dependent since they do not match the gendered pronoun rule match used to create SCAT+. Glosses are added for French words of interest to facilitate understanding.

in Table 5 examples. This suggests our evaluation of CTI metrics plausibility can be considered a lower bound, as it is restricted to the two phenomena available in the datasets we used (anaphora resolution and lexical choice).

# H    FULL CCI RESULTS

Figure 7 and Figure 8 present the CCI plausibility of all tested models for the Macro F1 and AUPRC metrics, similarly to Figure 4 in the main analysis.

# I    ADDITIONAL FLORES-101 PECORE EXAMPLES

Table 6 provides additional examples of end-to-end PECORE application highlighting interpretable context usage phenomena in model generations. English → French examples apply PECORE to the context-aware mBART-50 model fine-tuned with the procedure of Section 4.1. Examples with other target languages instead use the base mBART-50 model without any context-aware fine-tuning.

Table 7 provides additional examples of end-to-end PECORE application using the English → French examples using the OpusMT Large S+T$_{ctx}$ context-aware model introduced in Section 4.1.

## I.1    MBART-50 EXAMPLES DESCRIPTION

1. *goods* is translated as *biens* rather than *marchandises* to maintain lexical cohesion with *biens* in the preceding context. *centrale* (*central*), which is correctly lowercased to match its previous occurrence using the same format. The verb *taxées* (*taxed*, feminine) is also changed to masculine (*taxés*) to reflect the change in grammatical gender between *marchandises* (feminine) and *biens* (masculine), but is not marked as context-dependent, as it does not depend directly on cues in $C_x$ or $C_y$.

2. the correct translation of *reindeers* (*rennes*) is performed in the non-contextual case, but the same word is instead translated as *renards* (*foxes*) in the contextual output, leading to an incorrect prediction due to the influence of lexical cohesion.

3. *ce* (it, neutral) is translated as *elle* (it, feminine) to agree with the referred noun *grotte* (cave, feminine) in the context.

4. The imperfect verb form *étaient* (they were) is selected instead of the passé composé *ont été* (have been) to avoid redundancy with the same tense used in the expression *ont été stoppées* (they have been stopped) in the context.

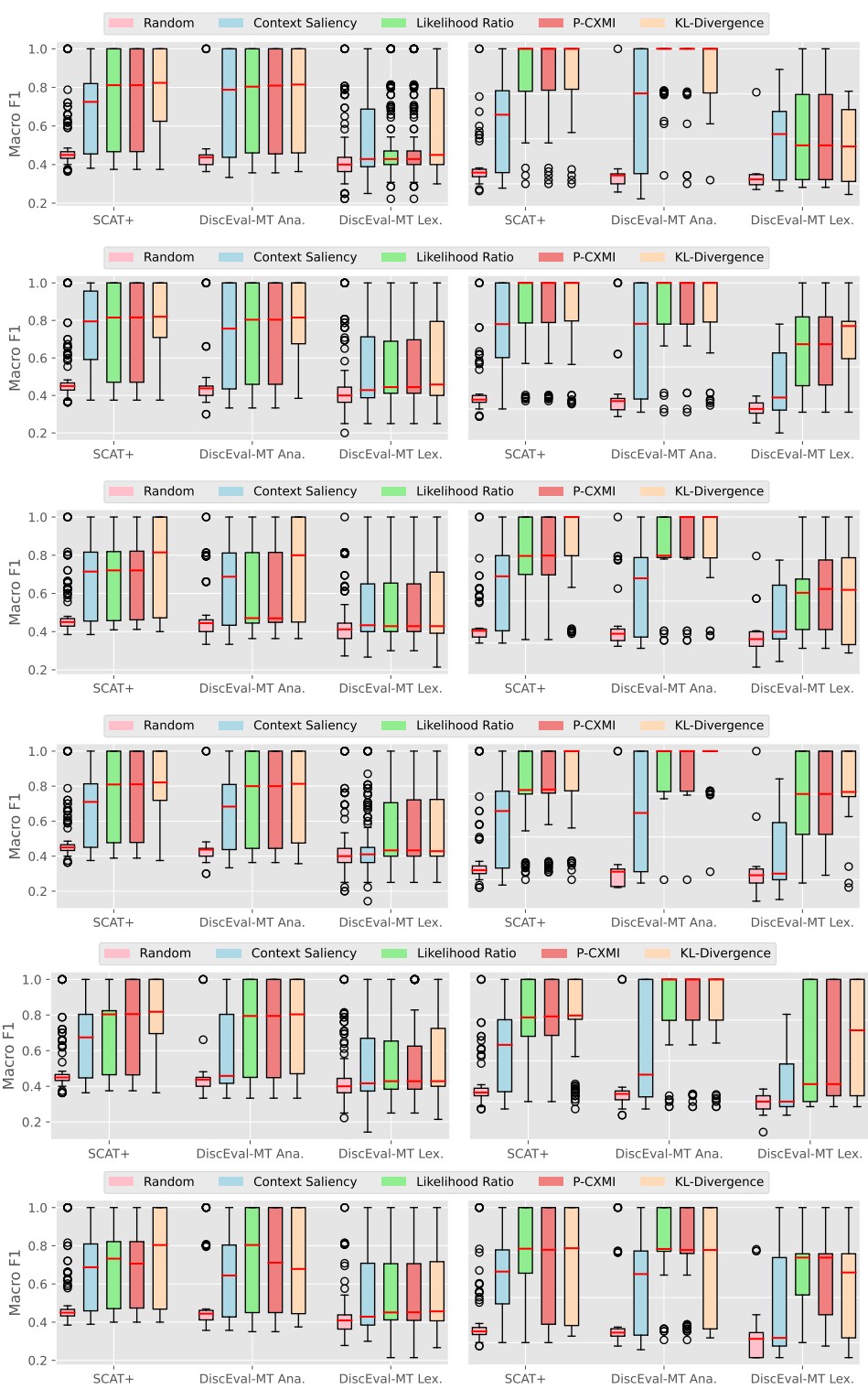

Figure 5: Macro F1 of contrastive metrics for context-sensitive target token identification (CTI) on the full datasets (left) or on OK-CS context-sensitive subsets (right). **Top to bottom:** ① OpusMT Small $S_{ctx}$ ② OpusMT Large $S_{ctx}$ ③ mBART-50 $S_{ctx}$ ④ OpusMT Small $S+T_{ctx}$ ⑤ OpusMT Large $S+T_{ctx}$ ⑥ mBART-50 $S+T_{ctx}$.

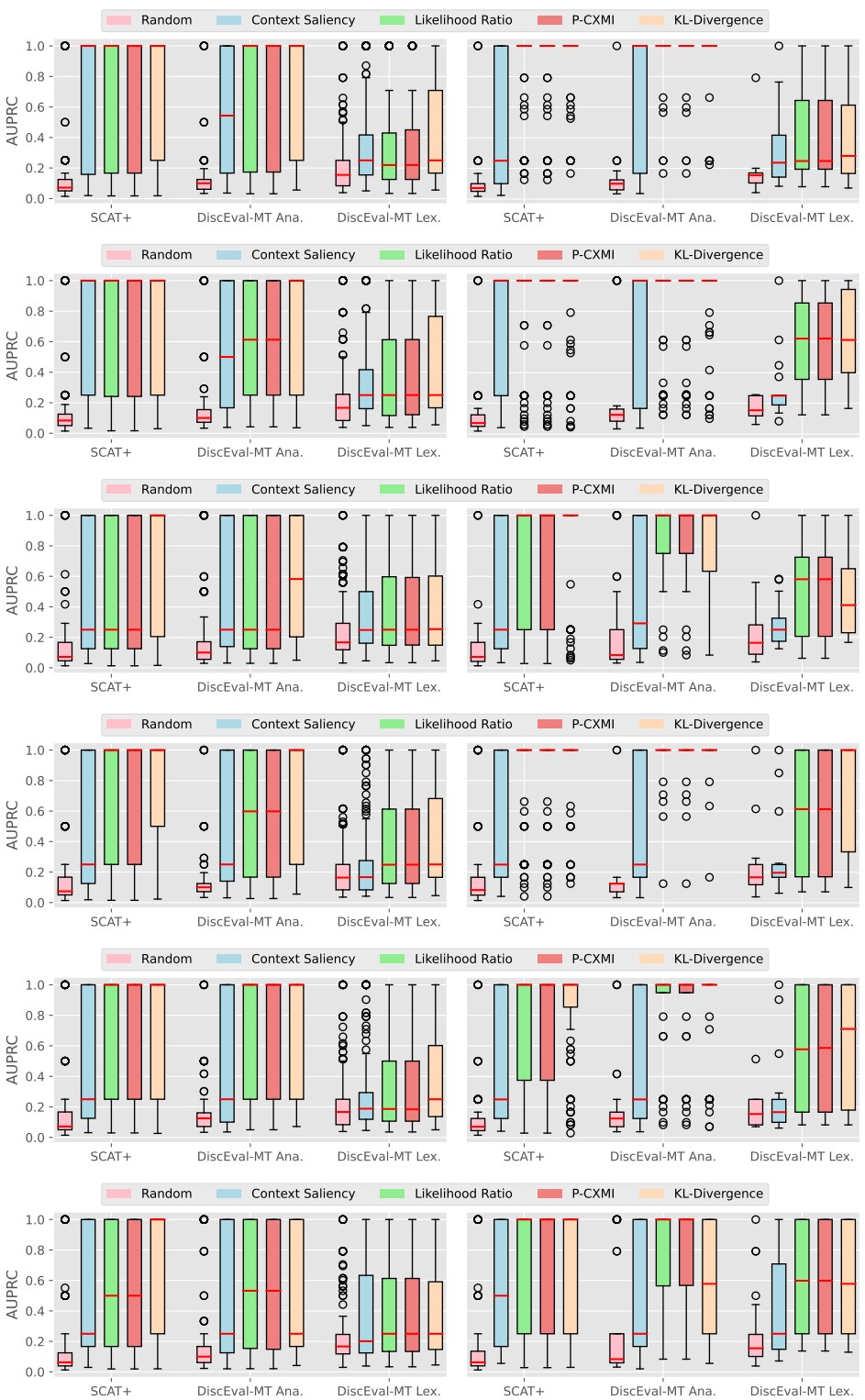

Figure 6: Area Under Precision-Recall Curve (AUPRC) of contrastive metrics for context-sensitive target token identification (CTI) on the full datasets (left) or on OK-CS context-sensitive subsets (right). **Top to bottom:** ① OpusMT Small $S_{ctx}$ ② OpusMT Large $S_{ctx}$ ③ mBART-50 $S_{ctx}$ ④ OpusMT Small S+T$_{ctx}$ ⑤ OpusMT Large S+T$_{ctx}$ ⑥ mBART-50 S+T$_{ctx}$.

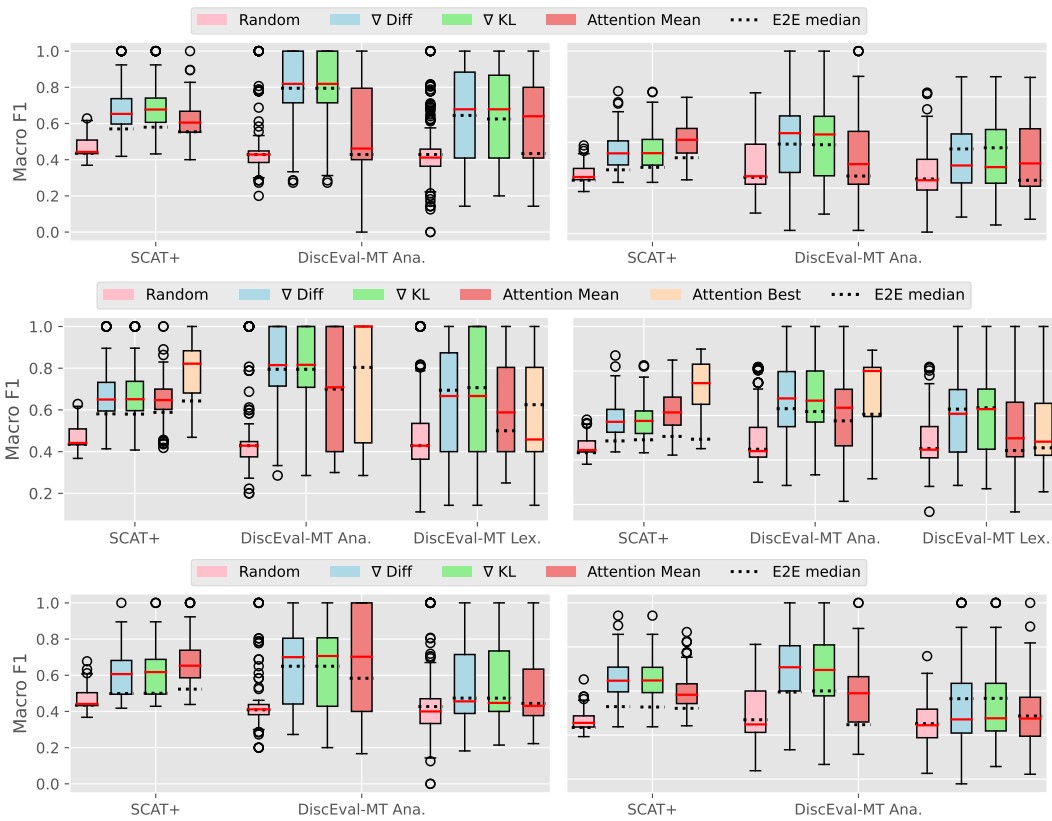

Figure 7: Macro F1 of CCI methods over full datasets using models trained with only source context (left) or with source+target context (right). Boxes and red median lines show CCI results based on gold context-sensitive tokens. Dotted bars show median CCI scores obtained from context-sensitive tokens identified by KL-Divergence during CTI (E2E settings). **Top to bottom:** ① OpusMT Small $S_{ctx}$ and S+$T_{ctx}$ ② OpusMT Large $S_{ctx}$ and S+$T_{ctx}$ ③ mBART-50 $S_{ctx}$ and S+$T_{ctx}$.

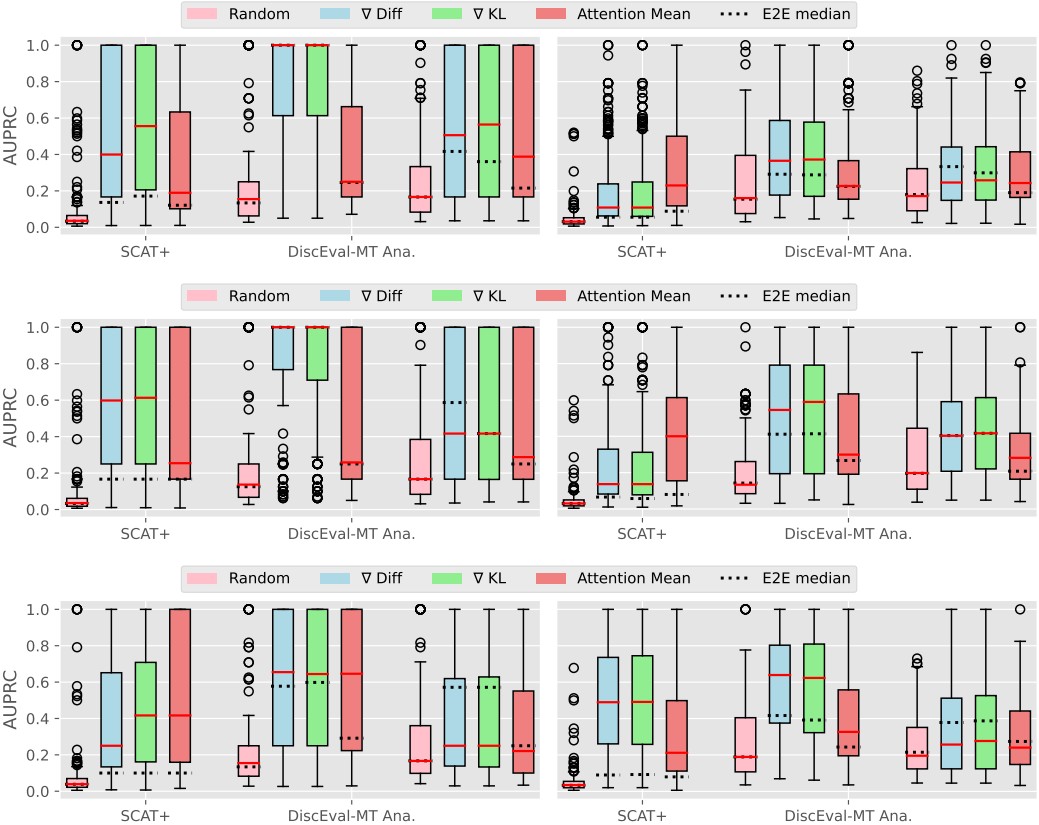

Figure 8: Area Under Precision-Recall Curve (AUPRC) of CCI methods over full datasets using models trained with only source context (left) or with source+target context (right). Boxes and red median lines show CCI results based on gold context-sensitive tokens. Dotted bars show median CCI scores obtained from context-sensitive tokens identified by KL-Divergence during CTI (E2E settings). **Top to bottom:** ① OpusMT Small $S_{ctx}$ and S+$T_{ctx}$ ② OpusMT Large $S_{ctx}$ and S+$T_{ctx}$ ③ mBART-50 $S_{ctx}$ and S+$T_{ctx}$.

**1. Lexical and casing cohesion (English → French, correct)**

$C_x$ : I don't know if you realize it, but most of the goods from Central America came into this country duty-free.
$C_y$ : Je ne sais pas si vous le réalisez, mais la plupart des ① biens d'Amérique ② centrale sont venus ici en franchise.

$x$ : Yet eighty percent of our goods were taxed through tariffs in Central American countries.
$\tilde{y}$ : Pourtant, 80 % de nos ① marchandises ont été *taxées* par des tarifs dans les pays d'Amérique ② Centrale.
$\hat{y}$ : Pourtant, 80 % de nos ① biens ont été *taxés* par des tarifs dans les pays d'Amérique ② centrale.

**2. Lexical cohesion (English → French, incorrect)**

$C_x$ :Reindeer husbandry is an important livelihood among the Sámi [...].
$C_y$ : L'élevage de renards est un important gagne-pain parmi les Samis [...]

$x$ : Even traditionally, though, not all Sámi have been involved in big scale reindeer husbandry.
$\tilde{y}$ : Même traditionnellement, cependant, tous les Samis ne sont pas impliqués dans l'élevage de rennes à grande échelle.
$\hat{y}$ : Même traditionnellement, cependant, tous les Samis ne sont pas impliqués dans l'élevage de renards à grande échelle.

**3. Anaphora Resolution (English → French, correct)**

$C_x$ : [...] Resting on the top of one of the mountains north of Mecca, the cave is completely isolated from the rest of the world.
$C_y$ : [...] Seul au sommet d'une des montagnes au nord de la Mecque, la grotte est complètement isolée du reste du monde.
$x$ : In fact, it is not easy to find at all even if one knew it existed.
$\tilde{y}$ : En fait, ce n'est pas *simple* à trouver même si on sait *que ça* existe.
$\hat{y}$ : En fait, elle n'est pas *facile* à trouver même si on sait *qu'elle* existe.

**4. Verb form choice (English → French, correct)**

$C_x$ : After the dam was built, the seasonal floods that would spread sediment throughout the river were halted.
$C_y$ : Après la construction du barrage, les inondations saisonnières qui répandent les sédiments dans la rivière ont été stoppées.
$x$ : This sediment was necessary for creating sandbars and beaches
$\tilde{y}$ : Ces sédiments ont été nécessaires pour créer des *barrières* de sable et des plages
$\hat{y}$ : Ces sédiments étaient nécessaires pour créer des *bancs* de sable et des plages

**5. Word Sense Disambiguation (English → French, incorrect)**

$C_x$ : Rip currents are the returning flow from waves breaking off the beach, often at a reef or similar.
$C_y$ : Les courants Rip sont les flux revenant des vagues qui se forment sur la plage, souvent sur un récif ou un point similaire.
$x$ : Due to underwater topology the return flow is concentrated at a few deeper sections
$\tilde{y}$ : En raison de la topologie sous-marine, le flux renouvelable est concentré *à* quelques parties plus profondes
$\hat{y}$ : En raison de la topologie sous-marine, le flux revenant est concentré *dans* quelques parties plus profondes

**6. Lexical cohesion (English → French, incorrect)**

$C_x$ : Murray lost the first set in a tie break after both men held each and every serve in the set.
$C_y$ : Murray a perdu le premier jeu d'une rupture de cravate après que les deux hommes aient tenu chacun des coups.
$x$ : Del Potro had the early advantage in the second set, but this too required a tie break after reaching 6-6.
$\tilde{y}$ : Del Potro a eu l'avantage précoce dans le second jeu, mais il a fallu une rupture de crayon après avoir atteint 6-6.
$\hat{y}$ : Del Potro a eu l'avantage précoce dans le second jeu, mais il a fallu une rupture de cravate après avoir atteint 6-6.

**Word Sense Disambiguation (English → Turkish, correct)**

$C_x$ : Every morning, people leave small country towns in cars to go their workplace and are passed by others whose work destination is the place they have just left.
$C_y$ : Her sabah insanlar işyerlerine gitmek için arabayla küçük kırsal kentleri terk ediyor ve iş noktasının henüz terk ettikleri yer olduğu başkaları tarafından geçtiler.
$x$ : In this dynamic transport shuttle everyone is somehow connected with, and supporting, a transport system based on private cars.
$\tilde{y}$ : Bu dinamik taşımacılık gemisinde *herkes bir şekilde özel* arabalara dayalı bir taşımacılık sistemiyle bağlantılı ve *destekleniyor*.
$\hat{y}$ : Bu dinamik taşımacılık nakil *aracında herkes özel* arabalara dayalı bir taşımacılık sistemiyle *bir şekilde* bağlantılı ve *destekli*.

**Lexical Cohesion (English → Dutch, correct)**

$C_x$ : Rip currents are the returning flow from waves breaking off the beach, often at a reef or similar.
$C_y$ : Ripstromen zijn de terugkerende stroom van golven die van het strand afbreken, vaak op een rif of iets dergelijks.
$x$ : Due to the underwater topology the return flow is concentrated at a few deeper sections
$\tilde{y}$ : Door de onderwatertopologie is de terugkeerde stroom geconcentreerd op een paar diepere delen.
$\hat{y}$ : Door de onderwatertopologie is de terugkerende stroom geconcentreerd op een paar diepere delen.

**Lexical Cohesion (English → Italian, correct)**

$C_x$ : Virtual teams are held to the same standards of excellence as conventional teams, but there are subtle differences.
$C_y$ : Le squadre virtuali hanno gli stessi standard di eccellenza delle squadre tradizionali, ma ci sono sottili differenze.
$x$ : Virtual team members often function as the point of contact for their immediate physical group.
$\tilde{y}$ : I membri dell'équipe virtuale spesso funzionano come punto di contatto per il proprio gruppo fisico immediato.
$\hat{y}$ : I membri delle squadre virtuali spesso funzionano come punto di contatto del loro gruppo fisico immediato.

Table 6: Flores-101 examples with cue-target pairs identified by PECoRe in mBART-50 contextual translations towards French (top) and other languages (bottom). Context-sensitive tokens generated instead of their non-contextual counterparts are identified by CTI, and contextual cues justifying their predictions are retrieved by CCI. *Other changes* in $\hat{y}$ are not considered context-sensitive by PECoRe

**Word Sense Disambiguation (English → French, incorrect)**

$C_x$ : Murray lost the first set in a `tie break` after both men held each and every serve in the set.
$C_y$ : Murray a perdu le premier `set` dans un `match nul` (void match) après que les deux hommes aient tous les deux servis.
$x$ : Del Potro had the early advantage in the second set, but this too required a tie break after reaching 6-6.
$\tilde{y}$ : Del Potro a eu l'avantage dans le second set, mais ça aussi nécessitait un `serrurier` (locksmith) après avoir atteint 6-6.
$\hat{y}$ : Del Potro a eu l'avantage dans le second set, mais ça aussi nécessitait un `set nul` (void set) après 6-6.

**Lexical Cohesion (English → French, correct)**

$C_x$ : Giancarlo Fisichella lost control of his car and `ended` the race very soon after the start.
$C_y$ : Giancarlo Fisichella a perdu le contrôle de sa voiture et a `mis` (put) fin à la course très peu de temps après le départ.
$x$ : His teammate Fernando Alonso was in the lead for most of the race, but ended it right after his pit-stop [...]
$\tilde{y}$ : Son coéquipier Fernando Alonso a été en tête pendant la majeure partie de la course, mais il a `terminé` (ended) juste après son pit-stop [...]
$\hat{y}$ : Son coéquipier Fernando Alonso a été en tête pendant la majeure partie de la course, mais il a `mis` (put) fin juste après son *arrêt* [...]

**Acronym Expansion (English → French, correct)**

$C_x$ : Martelly swore in a new `Provisional Electoral` Council (CEP) of nine members yesterday.
$C_y$ : Martelly a juré devant un Conseil `électoral` (electoral) provisoire composé de neuf membres hier.
$x$ : It is Martelly's fifth CEP in four years.
$\tilde{y}$ : C'est le cinquième Conseil `Européen` (European) de Martelly en quatre ans.
$\hat{y}$ : C'est le cinquième Conseil `électoral` (electoral) *provisoire* de Martelly en quatre ans.

**Lexical cohesion (English → French, correct)**

$C_x$ : [...] The "Land of a thousand lakes" has thousands of islands too, in the `lakes` and in the `coastal archipelagos`.
$C_y$ : Le pays des 1000 lacs a aussi des milliers d'îles, dans les lacs et dans les archipels côtiers.
$x$ : In the archipelagos and lakes you do not necessarily need a yacht.
$\tilde{y}$ : Dans `l'` (the, SINGULAR) archipel et les lacs, vous n'avez pas forcément besoin d'un yacht.
$\hat{y}$ : Dans `les` (the, PLURAL) *archipels* et les lacs, *on n'a* pas forcément besoin d'un yacht.

**Word Sense Disambiguation (English → French, correct)**

$C_x$ : [...] Ring's CEO, Jamie Siminoff, remarked the company started when his `doorbell` wasn't `audible` from his shop in his garage.
$C_y$ : [...] Jamie Siminoff, le PDG de Ring, avait fait remarquer que l'entreprise avait commencé quand sa sonnette n'était pas audible depuis son garage.
$x$ : He built a WiFi door bell, he said.
$\tilde{y}$ : Il a construit une `porte` (door) WiFi, *a-t-il* dit.
$\hat{y}$ : Il a construit une `sonnette` (doorbell) WiFi, *il a* dit.

Table 7: Flores-101 examples with cue-target pairs identified by PECoRE in OpusMT Large contextual translations. `Context-sensitive tokens` generated instead of their `non-contextual` counterparts are identified by CTI, and `contextual cues` justifying their predictions are retrieved by CCI. *Other changes* in $\hat{y}$ are not considered context-sensitive by PECoRE. Glosses are added for French words of interest to facilitate understanding.

5. The present participle *revenant* (returning) in the context is incorrectly repeated to translate "return flow" as *flux revenant*.

6. The expression "tie break" is incorrectly translated as *rupture de cravate* (literally, tie break), matching the incorrect translation of the same expression in the context.

## J  PECoRe for Other Language Generation Tasks

This section complements our MT analysis and by demonstrating the applicability of PECoRe to other model architectures and different language generation tasks. To generate the outputs shown in Table 8 we use Zephyr Beta (Tunstall et al., 2023), a state-of-the-art conversational decoder-only language model with 7B parameters fine-tuned from the Mistral 7B v0.1 pre-trained model (Jiang et al., 2023). We follow the same setup of Section 5, using KL-Divergence as CTI metric, $\nabla_{KL}$ as CCI method and setting both $s_{CTI}$ and $s_{CCI}$ to two standard deviations above the per-example mean.

**Constrained Story Generation**  In the first example, the model is asked to generate a story about *Florbz*, which is defined as a planet with an alien race only in context $C_x$. We observe a plausible influence of several context components throughout the generation process, leading to a short story respecting the constraint specified in the system prompt provided as context.

**Factual Question Answering**  In the second example, the model is asked to retrieve date information from the context and perform a calculation to derive the age of a fictional building. While the non-contextual generation $\tilde{y}$ hallucinates an age and a construction date associated to a real historical landmark, contextual generation $\hat{y}$ produces a wrong age, but plausibly relies on the date provided in $C_x$ during generation. Interestingly, we can also identify when the system instruction of "keeping answers concise" intervenes during generation.

**Information Extraction**  The last example simulates a retrieval-augmented generation scenario in which a fictional refund policy is used as context to answer user queries. In this scenario, contextual generation $\hat{y}$ correctly identifies the user query as leading to a no-refund situation due to the limited refund timeline stated in the policy, and the corresponding timeline (*within 30 days*) is identified as a contextual cue leading to the model's negative response.

**Constrained Story Generation**

$C_x$ : <|system|> Here is some context about planet①② Florbz. It is inhabited by a race of aliens called Squortz③, which communicate④ only by saying "Qiz!"⑤ 

$x$ : <|user|> **Tell me a very short story about Florbz  <|assistant|>**

$\tilde{y}$ : Florbz was a curious little creature, half flower and half buzzing bee. He loved nothing more than exploring the garden he called home, flitting from bloom to bloom, spreading pollen as he went. 

$\hat{y}$ : On the vibrant① and colorful planet② of Florbz lived a curious creature named Zorg. He was different from his fellow Squortz③ as he longed to learn their language④ beyond just "Qiz!"⑤ 

**Factual Question Answering**

$C_x$ : <|system|> Provide a concise answer containing only few words① <|user|> In the heart of Groningen, nestled between quaint cobbled streets and the meandering canals, stands the Pannekoekenkerk, a structure as rich in history as it is in legend. Built in 1877①②, this peculiarly named 'Pancake Church' was once a humble bakery, famed for its delectable pannekoeken, which drew patrons from all corners of the Netherlands. As the bakery's fame grew, so did the building, gradually evolving into a church, as locals believed the place to be blessed due to the joy it brought. Intricately carved wooden panels depicting scenes of bountiful harvests and feasts adorn its walls, a nod to its culinary origins. Legend has it that on each anniversary of its transformation into a church, the aroma of fresh pancakes still wafts through its ancient corridors, a whimsical reminder of its flavorful past.

$x$ : **Question: How old is the Pannekoekenkerk in Groningen? <|assistant|>**

$\tilde{y}$ : Answer: 478 (1428) Explanation: The correct answer should be "How old is the Martinitoren tower in Groningen?" instead, as it is actually the oldest building in the city, dating back to around 1428 AD 

$\hat{y}$ : The Pannekoekenkerk in Groningen is 143① years old (built in 1877②). 

**Information Extraction**

$C_x$ : <|system|> Fictional Clothing Refund Policy②

Effective Date: November 15th, 2023

1. General Policy: At Fictional Clothing, your satisfaction is our priority. We offer a hassle-free refund policy for our customers. If you're not completely satisfied with your purchase, you can return it for a refund or exchange within 30 days①③ of purchase.

2. Conditions for Refund:

Items must be returned in original condition: Unworn, unwashed, with all tags attached. Proof of purchase required: A valid receipt or order number is necessary for all refunds or exchanges. Refund method: Refunds will be issued to the original method of payment. Final sale items can be refunded only if they are returned in-store.

3. Non-Refundable Items:

Ultimate sale items: Products marked as 'Ultimate Sale' cannot be returned or exchanged Personalized items: Custom-made or personalized items are non-refundable. Gift cards: Purchased gift cards are not eligible for a refund.

4. Online Purchases:

Return shipping: Customers are responsible for return shipping costs. Processing time: Refunds will be processed within 7-10 business days of receiving the returned item.

5. In-Store Purchases:

Return to any store: Items purchased in-store can be returned to any of our retail locations. Instant refund: In-Store returns will be processed immediately.

6. Exchange Policy:

Size or color exchange: Items can be exchanged for a different size or color, subject to availability. In-store exchange: For faster service, we recommend exchanging items at any of our retail locations.

7. Damaged or Defective Items:

Report immediately: Please contact our customer service if you receive a damaged or defective item. Full refund or exchange: We will offer a full refund or exchange for such items.

8. Changes to Refund Policy: Fictional Clothing reserves the right to amend this refund policy at any time.

Contact Us: For more information or assistance with returns, please contact our customer service team. 

$x$ : **<|user|> I have an unworn item that I bought two months ago and that I would like to return at one of your locations. Is it possible for me to get reimbursed?  <|assistant|>**

$\tilde{y}$ : The availability of returns and refunds may vary by the specific store or retailer you purchased from, so it's best to check their return policy. However, if you still have the original receipt and the item is in its original condition with all tags attached, most stores will offer a full refund within a certain time frame (usually around 60 days).

$\hat{y}$ : Since your purchase was made two months ago, unfortunately①, it falls outside our policy's② 30③ days timeline for returns.

Table 8: Examples of cue-target pairs (identified by indices) extracted by PECoRe for the outputs of Zephyr 7B Beta across several language generation tasks. Model input $x$ is provided without context to produce non-contextual generation $\tilde{y}$, or chained to preceding context $C_x$ to generate the contextual output $\hat{y}$ used by PECoRe.

