# OpenReview forum: "Quantifying the Plausibility of Context Reliance in Neural Machine Translation"
_ICLR.cc/2024/Conference — ICLR 2024 poster_

### Official Review · Reviewer_GSke · 2023-11-02

**Soundness:** 3 good
**Presentation:** 2 fair
**Contribution:** 3 good
**Rating:** 6
**Confidence:** 2

**Summary:**

This work proposes an evaluation method for context aware machine translation to quantify how a model is looking at words in the input for their decision for translation. Basically, the evaluation is carried out in two steps, the first step (CTI) to detect the target tokens which are sensitive to context information by differentiating a model with and without context, and the second step (CCI) to perform attribution to the input tokens using the detected target tokens.

**Strengths:**

- This work is investigating an important problem in context aware machine translation evaluation in that a context aware model usually ignore important signals in the context by explain-away effects. The evaluation protocol presented in this paper might be a way to give us an insight to a particular problem in a model, and thus, might lead to a solution in the future.

- The proposed method is a combination of two, one to isolate the particular tokens for context awareness by comparing two systems, and the other for attribution to find the import inputs to investigate the connection. It simple but sound, and easy to interpret the outputs.

**Weaknesses:**

- Experiments are not very systematic in that OpenMT was mainly employed in section 4 for evaluating each step, but section 5 used mBART-50 in an end-to-end evaluation setting. I'd expect two settings in the end-to-end experiments for completeness.

**Questions:**

- I'd like to know the reason for not running OpenMT for the end-to-end experiment in section 5.

---

> ### Author Response · Authors · 2023-11-16
> **Author Response to Reviewer GSke**
>
> Dear Reviewer GSke,
>
> Thank you for taking the time to actively engage with our submission! We are glad you found our method “simple but sound, with easy-to-interpret outputs,” and appreciate your comment about our work addressing “an important problem in context-aware machine translation.”
>
> **Usage of mBART in Section 5 end-to-end experiments**
>
> Our initial decision to use mBART for our end-to-end analysis in Section 5 was aimed at showcasing the applicability of our approach to other models and target languages. However, we agree with your comment suggesting that missing OpusMT examples might undermine the systematicity and completeness of our analysis since the system was used in previous sections. For this reason, in the revised version of our paper, we extend our end-to-end analysis to the OpusMT Large model, adding two examples of OpusMT results in Section 5 and several more in Appendix I (Table 7). We observe that OpusMT results present several issues with lexical cohesion that closely match those previously shown for mBART, making the new result consistent with our original analysis.

---

### Official Review · Reviewer_DAmX · 2023-11-02

**Soundness:** 2 fair
**Presentation:** 1 poor
**Contribution:** 2 fair
**Rating:** 3
**Confidence:** 2

**Summary:**

It is an important topic to evaluate if NLP models can correctly make use of context information in a plausible way, which is called plausibility. It is widely studied in classification tasks but relatively challenging for language generation tasks. Previous works confines on the assessment to a small, handcrafted benchmarks. In this paper, the authors propose an interpretability framework called PECoRE. PECoRe extracts cue-target token pairs, which are used to uncover context dependence in generation. Comparing the uncovered dependence with human annotation, PECoRE can quantify the plausibility of language models. The authors apply PECoRE to evaluate contextual machine translation tasks, using the metrics or context sensitive target token identification and contextual cues imputation, as well as detecting context reliance in the wild, showing the effectiveness of PECoRE.

**Strengths:**

1. The topic (whether NLP models can correctly make use of context information in a plausible way) is important for trustworthy AI.

**Weaknesses:**

1.	This paper is hard to read. The logic and the main message of each paragraph are hard to follow. The writing is not friendly for the reader that does not familiar to plausibility study. Also, the examples are not hard to understand for the reader that cannot understand French.
2.	The scope is narrow. The proposed interpretability method seems only suitable for encoder-decoder based machine translation models. But the focus of the language generation field has move to more general decoder-based language generation, such as GPT-style.
3.	Lack of comparation. The proposed method are not compared with other SOTA methods for plausibility.

**Questions:**

1.	Although your method is for generative language models, it seems that it is more suitable for seq2seq generation tasks, such as machine translation, rather than pure generation tasks, such as story generation. Also, it seems only works for encoder-decoder structure, rather than currently widely used decoder-only language model. Can your method work in such scenarios?

2.	Where is the comparison between your method and other SOTA method? Correct me if I am wrong, it seems that you only evaluate the performance of different metrics on different models.

---

> ### Author Response · Authors · 2023-11-16
> **Author Response to Reviewer DAmX**
>
> Dear Reviewer DAmX,
>
> Thank you for taking the time to engage actively with our submission! We are glad you consider the topic of our research “important for trustworthy AI.”
>
> **Hard-to-read with unclear takeaways, plausibility-specific terminology and French**
>
> - We have revised remarks in Section 3.2 and results in Sections 4.3 and 4.5 to make the logic and conclusions clearer.
> - We have added an appendix section (Appendix A) to introduce some terminology adopted in plausibility studies, provide a concrete example of current practices in plausibility evaluation, and clarify the functioning of our adopted plausibility metrics for people less familiar with this research domain.
> - While it is inevitable to depend on non-English examples for a study focusing on machine translation, we added English glosses to Section 5 examples and appendix tables to simplify understanding of relevant elements in model translations.
>
> **Narrow scope, encoder-decoder specific method**
>
> The PECoRe framework is not architecture-specific and can be applied to modern decoder-only large language models. We apologize for not making this clearer in the initial version of the paper, and we have added a clarifying remark in the revised version of the introduction. Moreover, we have added some examples in Appendix J to further highlight the broad applicability of our approach to study the plausibility of a state-of-the-art decoder-only large language model to quantify context reliance in question answering, story generation tasks and information extraction. Our preliminary evidence suggests that PECoRe can be useful to ensure a trustworthy context reliance in these domains.
>
> **Lack of comparisons to established practices in plausibility evaluation**
>
> The established practice for plausibility evaluation is to verify the overlap between model rationales and human annotation for pre-defined target tokens of interest. Our framework, instead, proposes a more comprehensive evaluation framework that includes the identification of those targets. The comparison between the two is performed in Section 4.5. The CCI results from gold context-sensitive tokens in Figure 4 reflect the traditional hypothesis-driven procedure for plausibility evaluation. Additional background and a concrete example of plausibility evaluation are now provided in Appendix A to avoid any confusion.
>
> Instead, end-to-end medians in the same Figure represent the outcome of our proposed framework, leading to more conservative plausibility estimates that reflect a broader notion of model contextual reliance, incorporating the detection of context-sensitive target tokens in the evaluation.
>
> Importantly, the goal of our work is not to adopt the traditional “SOTA” procedure for plausibility evaluation to improve results on some metrics of interest (e.g. Macro F1 between human and model rationales in Figure 4), but to address the shortcomings of this procedure by proposing a shift from hypothesis-driven to data-driven plausibility evaluation. This shift ultimately enables its usage in real-world settings where references and annotations are unavailable. We added a sentence in the revised Section 4.5 to clarify this aspect.
>
> In terms of our choice of methods, the P-CXMI method can be seen as the “previous SOTA” for CTI, as it was used by [1] for context sensitivity detection, while for CCI, the Attention Mean method by [2]  was the best one that does not require pre-hoc knowledge (as in the case of Attention Best) before conducting the evaluation.
>
> **References**
>
> [1] [When Does Translation Require Context? A Data-driven, Multilingual Exploration](https://aclanthology.org/2023.acl-long.36) (Fernandes et al., ACL 2023)
>
> [2] [When and Why is Document-level Context Useful in Neural Machine Translation?](https://aclanthology.org/D19-6503) (Kim et al., DiscoMT 2019)

---

### Official Review · Reviewer_kdfb · 2023-11-02

**Soundness:** 4 excellent
**Presentation:** 4 excellent
**Contribution:** 3 good
**Rating:** 8
**Confidence:** 4

**Summary:**

The paper introduces PECORE, a novel interpretability framework for analyzing context usage in language models' generations. For any (contextual) generation model, PECORE proposes extracting sets of (target, cue) corresponding to what parts of the generation were reliant on context, and what parts of the context they relied on.

The framework has two main steps:

1. Context-sensitive target identification (CTI): This step identifies which tokens in the model's generated text were likely influenced by the preceding context. It does this by comparing the model's predictions with and without context using contrastive metrics.
2. Contextual cues imputation (CCI): This step traces the generation of context-sensitive tokens (for “gold” context-sensitive words or end-to-end from the CTI step) back to specific words in the input context that influenced their prediction. It uses attribution methods to identify influential contextual cues.

The (target,cue) outputs can then be compared to human rationales to evaluate the *plausability*. Importantly, the PECORE framework is agnostic to the specific method used in the CTI step and CCI step, and authors investigate the plausability of PECORE using different combinations of methods.

The authors apply PECORE to specific task of contextual machine translation. Experiments on discourse phenomena datasets show the PECORE can identify context-sensitive outputs and trace them back to contextual cues with reasonable accuracy (when compared to human-rationales), particularly when using simple distributional metrics on both steps (like CXMI or KL-divergence). The framework is also applied to unannotated examples in FLORES dataset, revealing interesting cases of context usage in translations.

**Strengths:**

- I really liked this paper! It is well written and the presentation is quite clear, and the problem of interpretability/context-attribution is understudied given how relevant it is. They also show understanding of the current problems in the interpretability literature, by explicitly mentioning that they only measuring plausibility (similarity to human rationales)
- I find their CTI step quite ingenious: they side-step some of the problems of previous context-usage metrics that rely on references (like CXMI) by basically decoding with context and assuming that as “contextualized reference”. This means that they can use these “probabilistic” metrics to measure context on actual decodings/translations of the model rather than on references (which might not be representative of real use of the model).
- I find the comparison of different contrastive metrics and selectors through PECORE quite relevant, and a useful resource for future intepretability research. The finding that simple probabilistic metrics like CXMI/KL work well and that using the full-distribuion (like KL) can be beneficial quite relevant

**Weaknesses:**

- The main thing that could make this paper have a significantly bigger impact would be testing the use of PECORE for tasks other than contextual MT: as authors point out, PECORE is general enough to be applicable to any (contextual) LM, and some experiments on other tasks (like summarization or even other modalities) could make this even more suitable to conference like ICLR (doing something like what was done for FLORES if human rationales don’t exist).
- While I think the authors did well in side-stepping the issues with the faithfulness/plausibility argument, I think some parts of the writing still hint that these explanations reflect the model’s internal behaviour (e.g. “leaving space for improvement through the adoption of more faithful attribution” this implies that more faithful atribbution will lead to more plausible explanations, which might not be the case). Some extra care about this sort of statements could be useful (but overall I found that the authors were already quite careful).

**Questions:**

- Is it possible that $\tilde{y}^\star$ ends up being the same as the contextual prediction $\hat{y}$ (in remark 3.1)? Or does a high selector score imply that the highest probability token is gonna be different?

---

> ### Author Response · Authors · 2023-11-16
> **Author Response to Reviewer kdfb**
>
> Dear Reviewer kdfb,
>
> Thank you for taking the time to engage actively with our submission! We appreciate your comments stating that our paper is “well written and the presentation is quite clear” and “shows an understanding of the current problems in the interpretability literature.” We are especially happy that you find our proposed approach ingenious and our findings relevant to interpretability research.
>
> **Application of PECoRe to other tasks/modalities**
>
> As you correctly state, our framework can be applied to other tasks or modalities to study the impact of context dependence beyond context-aware MT. We agree that showcasing such applications would strengthen the presentation of our method, and for this reason, we include a new appendix section in the revised submission (Appendix J). This new section applies PECoRe to a state-of-the-art decoder-only large language model to quantify context reliance in question answering, story generation tasks and information extraction. Our preliminary evidence suggests that PECoRe can be useful to ensure a trustworthy context reliance in these domains.
>
> While such applications are undoubtedly exciting, we opt to include those as appendix materials since a more thorough evaluation of PECoRe for these tasks would, in our opinion, be more fitting for future studies leveraging task and modality-specific expertise to extend our investigation.
>
> **Statement conflating attribution plausibility with faithfulness**
>
> We agree that the sentence you report assumes a relation between faithfulness and plausibility when these two evaluation dimensions were shown to be largely independent in previous interpretability work. We have adjusted our formulation at the end of Section 4.5 to clarify that we are specifically interested in attribution methods that better align with human rationales.
>
> **Question regarding $\tilde y^{*}$ and $\hat y$**
>
> Even for cases where $\tilde y^{\*}$ is identical to $\hat y$, the probabilities associated with the tokens of the two sequences are likely to be different. Indeed, while both are produced from target prefix $\hat y_{<t}$, $\tilde y^{*}$ use non-contextual source inputs, while $\hat y$ uses contextual source inputs. If the probabilities of the same next token $\hat y_t$ are different, the result of the contrastive target function $f_{tgt}$ will also be different, leading to non-zero attribution scores over the context.
>
> The only case in which these scores would be equal to zero is when the probability of the next token (or the distribution over the vocabulary, for KL-Divergence) is exactly the same with and without input context, i.e. if context is not affecting the generation process. We have added a new remark (3.2) to clarify this aspect.

---

> > ### Comment · Reviewer_kdfb · 2023-11-20
> > **Response to Rebuttal**
> >
> > I would like to thank you to the reviewers. Given this and their (good) responses to the other reviews, I'm keeping my score.

---

### Official Review · Reviewer_sS3A · 2023-11-05

**Soundness:** 2 fair
**Presentation:** 3 good
**Contribution:** 3 good
**Rating:** 5
**Confidence:** 2

**Summary:**

This paper presents a comprehensive framework for assessing the plausibility of context reliance in the context of machine translation tasks. More specifically, this framework facilitates the extraction of cue-target token pairs from language model outputs, enabling the identification of context-sensitive target tokens and their associated influential contextual cues. The primary objective of this approach is to systematically evaluate the credibility of context utilization within the domain of machine translation, and notably, it can be applied in the absence of reference translations. Ultimately, the authors substantiate the framework's accuracy across diverse datasets and underscore its effectiveness

**Strengths:**

- The topic is very interesting
- A comprehensive explanation of the proposed algorithms is provided, along with supplementary materials for additional insight.
- The evaluation is encompassing a wide range of models.
- This model can be utilized without the need for reference.

**Weaknesses:**

- The authors introduce incremental methods, highlighting their proposed unified framework as a distinguishing feature compared to previous work[1]. The unified framework aims to enable an end-to-end evaluation of the plausibility of context reliance. However, it raises questions regarding the advantages of such unification. For instance, the Detection component (CTI) and CCI component could potentially function independently, and the potential synergy of combining them within a unified framework remains unclear. Additionally, each component closely resembles existing methodologies[1], with the experimentation primarily focused on various detection metrics. Notably, the paper's most advanced aspect, compared to existing methods, is its omission of reference translations, which may be perceived as somewhat lacking in novelty for an ICLR paper.

- The authors acknowledge limited performance, as indicated by the data presented in Table 1 and Figure 3. They attribute the suboptimal performance to domain shift issues affecting lexical choice. This suggests that the proposed methodology predominantly excels in handling anaphora cases. However, from my perspective, it's worth noting that lexical choice in context-aware translation is a significant aspect, and it's not entirely clear if the approach can effectively diagnose and interpret the model in practical context-aware translation research.

**Questions:**

- I find several typos in page 4
     - P_{c}tx (y^t)-> P_{ctx}(y^t)
     - P_{c}tx (y^*)-> P_{ctx}(y^*)

---

> ### Author Response · Authors · 2023-11-16
> **Author Response to Reviewer sS3A**
>
> Dear Reviewer sS3A,
>
> Thank you for actively engaging with our submission! We appreciate your comments mentioning that “the topic is very interesting” and that “the evaluation encompasses a wide range of models.” We are especially glad you found our explanation of the PECoRe approach comprehensive.
>
> **Novelty and advantages of a unified CTI+CCI framework**
>
> While it is true that CTI and CCI can be applied independently, we want to emphasize that PECoRe is not a simple sequential application of existing approaches. On the one hand, current evaluation practices use non-contrastive attribution methods as the only step to quantify model plausibility. In those, the hypothesis-driven pre-selection of attribution targets in a hypothesis-driven way inevitably biases the evaluation to a subset of known and expected behaviors (e.g. we added a clarifying example in Appendix A). On the other hand, CTI metrics were used for other purposes, such as filtering datasets of examples with context-dependent phenomena [1], and were not applied in the context of plausibility evaluation in previous work.
>
> Apart from selecting attribution targets in a data-driven way with CTI, PECoRe also uses CTI contextual and non-contextual outputs as targets for contrastive attribution, effectively harmonizing CTI inputs with CCI targets. To our knowledge, this is the first application of CTI methods and contrastive attribution for studying model plausibility, proposing a shift from hypothesis-driven to data-driven plausibility evaluation to enable its usage in real-world settings where references and annotations are unavailable (as we do in Section 5). Notably, [2] found the contrastive attribution approach we adopt for CCI to lead to improved simulatability for model decisions compared to regular attribution, providing an additional justification for our novel framework.
>
> **Limited performance and unclear generalizability to other contextual phenomena**
>
> Your statement, “the proposed methodology predominantly excels in handling anaphora cases,” implies that the weaker CTI scores on lexical choice in Figure 3 could be attributed to issues with our proposed PECoRe framework. However, this is not the case: Table 1 reports the performance of the MT model we use as a “test subject” and not the results of our proposed plausibility evaluation framework (Figures 3 and 4). These two aspects should not be conflated, as our method aims to detect cases of implausible model behavior and not to directly improve the performance of the underlying model.
>
> Indeed, from Table 1, it is evident that the MT model can less proficiently use contextual information for lexical choice, likely due to the limited presence of such phenomena in the training data. If the model cannot use contextual information to disambiguate lexical choices in several cases (Table 1, column “OK”), it is expected that CTI would identify this setting as implausible from a context-sensitivity perspective (Figure 3, Left). This motivates us to include results for the OK-CS split (Figure 3, right), the data subset for which we know context availability leads to the correct disambiguation. On this split, we can see that the proposed KL-Divergence metric fares competitively, suggesting the robustness of PECoRe despite the domain shift.
>
> As additional evidence of PECoRe’s generalizability to unseen phenomena, Appendix G (Table 5) in the revised draft presents some SCAT+ examples for whose a context sensitivity in the choice of grammatical formality (“you” → “toi/vous”) was correctly identified by the KL-Divergence CTI metric. However, those examples were considered wrong in the evaluation of Section 4.3 since it did not match the gender disambiguation phenomena (“it, they” → “il/elle, ils/elles”) used to annotate SCAT+.
>
> **Page 4 typos:** We thank you for reporting these typos. Following other reviewers’ questions, Remark 3.1 is adjusted, and a new Remark 3.2 is added in the revised version to further clarify the case in which the presence of input context leads to the same generations for $\tilde y^{\*}$.
>
> **References**
>
> [1] [When Does Translation Require Context? A Data-driven, Multilingual Exploration](https://aclanthology.org/2023.acl-long.36) (Fernandes et al., ACL 2023)
>
> [2] [Interpreting Language Models with Contrastive Explanations.](https://aclanthology.org/2022.emnlp-main.14) (Yin and Neubig, EMNLP 2022)

---

### Meta-Review · Area_Chair_iCsi · 2023-12-14

**Metareview:**

This work proposes an evaluation method (PECoRe) for context aware machine translation to quantify how a model is looking at words in the input for their decision for translation. The evaluation is carried out in two steps, the first step (CTI) detects the target tokens which are sensitive to context information by differentiating a model with and without context. The second step (CCI) performs attribution to the input tokens using the detected target tokens.
The topic (whether NLP models can correctly make use of context information in a plausible way) is an important topic. A comprehensive explanation of the proposed algorithms is provided, along with supplementary materials for additional insight. Compared to previous approaches, PECoRe does not need a reference translation which is a big plus.
There were some concerns that PECoRe is only a simple sequential application of existing CTI and CCI approaches. The authors addressed the concern by mentioning that other methods are not based on contrastive scoring. PECoRe is a shift from hypothesis driven (pre-selection of attribution targets) to data driven evaluation (contrastive scoring) and enables us to detect attributes beyond the ones specified in the hypothesis driven approaches.
There were a few more concerns from reviewers which were all clarified during the author's response.

**Justification For Why Not Higher Score:**

The paper only contains experimental results on one task (MT). This made it more focused towards a subset of the community and seems like a great fit for a poster presentation.

**Justification For Why Not Lower Score:**

As discussed above, the paper is interesting and there were no valid concerns raised form the reviewers. It is a great, novel contribution,

---

### Decision · Program_Chairs · 2024-01-16

Accept (poster)